# Kinetics of the ancestral carbon metabolism pathways in deep-branching bacteria and archaea

Tomonari Sumi [1,2]✉ & Kouji Harada [3,4]

The origin of life is believed to be chemoautotrophic, deriving all biomass components from carbon dioxide, and all energy from inorganic redox couples in the environment. The reductive tricarboxylic acid cycle (rTCA) and the Wood–Ljungdahl pathway (WL) have been recognized as the most ancient carbon fixation pathways. The rTCA of the chemolithotrophic *Thermosulfidibacter takaii*, which was recently demonstrated to take place via an unexpected reverse reaction of citrate synthase, was reproduced using a kinetic network model, and a competition between reductive and oxidative fluxes on rTCA due to an acetyl coenzyme A (ACOA) influx upon acetate uptake was revealed. Avoiding ACOA direct influx into rTCA from WL is, therefore, raised as a kinetically necessary condition to maintain a complete rTCA. This hypothesis was confirmed for deep-branching bacteria and archaea, and explains the kinetic factors governing elementary processes in carbon metabolism evolution from the last universal common ancestor.

[1] Research Institute for Interdisciplinary Science, Okayama University, 3-1-1 Tsushima-Naka, Kita-ku, Okayama 700-8530, Japan. [2] Department of Chemistry, Faculty of Science, Okayama University, 3-1-1 Tsushima-Naka, Kita-ku, Okayama 700-8530, Japan. [3] Department of Computer Science and Engineering, Toyohashi University of Technology, Tempaku-cho, Toyohashi 441-8580, Japan. [4] Center for IT-Based Education, Toyohashi University of Technology, Tempaku-cho, Toyohashi, Aichi 441-8580, Japan. ✉email: sumi@okayama-u.ac.jp

The concept of a last universal common ancestor (LUCA) has been attracting much interest among researchers on the early evolution and origin of life. Based on an extensive investigation of protein-coding genes in prokaryotic genomes, LUCA has been inferred to be an anaerobic chemoautotroph that uses the Wood–Ljungdahl (WL) pathway and exists in a hydrothermal setting[1]. Therefore, most of the organic carbon on early Earth was obtained by chemoautotrophic carbon fixation, thus sustaining life. Currently, there are seven known different autotrophic pathways responsible for carbon fixation, including the newly demonstrated reductive glycine (rGly) pathway[2]. These pathways are recognized to have partial overlaps, but the reasons for the redundancy and the forces of selection in their evolution remain unclear. The WL pathway, also called the reductive acetyl-CoA (coenzyme A) pathway, and the reductive tricarboxylic acid (rTCA) cycle are among the most ancient pathways responsible for the biosynthesis of the universal precursors of anabolism[3–6]. Even if we focus only on these ancient carbon metabolism pathways, diverse observations on the phylogenetic distribution of the network topology have not yet been given a universal and phylum-compatible interpretation.

The rTCA cycle is a reversal of the oxidative tricarboxylic acid (oTCA) cycle, which provides energy, reducing agents, and precursors for certain amino acids. However, citrate synthesis from acetyl-CoA (ACOA) and oxaloacetate, catalyzed by citrate synthase (CS), is regarded as one of the irreversible reactions in the oTCA cycle[7,8]. Therefore, in organisms using the rTCA cycle, CS has been interpreted to be substituted either by an adenosine triphosphate (ATP)-dependent citrate lyase or other homologous enzymes that catalyze the same reaction in two steps[9–11]. In contrast, however, two research groups demonstrated that CS activity drove the rTCA cycle in the thermophilic bacteria *Thermosulfidibacter takaii* and *Desulfurella acetivorans* grown chemolithoautotrophically[12–14].

In the present study, kinetic simulations of the autotrophic rTCA cycle for these bacteria are presented and the mechanism by which the CS reaction is reversed, which was thought to function only in the oxidative direction, is revealed in dynamical systems. In addition, bifurcation of the TCA cycle into reductive and oxidative reactions that has been demonstrated in the presence of succinate and acetate[13,14] is also simulated. Interestingly, it was observed in dynamical systems that a conflict between the reductive and oxidative fluxes in the TCA cycle was caused by an influx of ACOA upon uptake of acetate. This indicates that the coexistence of a WL pathway and a rTCA cycle, which yields ACOA influx into the rTCA cycle, results in a partial reduction of fluxes in the TCA cycle. Consequently, the less active enzymes of the rTCA cycle due to this coexistence would be lost to evolution or enzymes lacked in the rTCA cycle would be difficult to be newly incorporated into the less active part of that. In the pioneering work by Braakman and Smith[5], a single connected redundant network consisting of a complete WL pathway and a full rTCA cycle was proposed as a more robust and plausible topology that LUCA or (proto) biological systems leading up to the LUCA possibly possess. However, our kinetic results do not support the coexistence of these two pathways in one organism, unless other predominant selective pressure that favors their combination exists as well. Remarkably, it was confirmed using the Kyoto Encyclopedia of Genes and Genomes (KEGG) carbon metabolism database[15] that our kinetic hypothesis, that a complete rTCA cycle does not coexist with a WL pathway, was held for deeply branching archaea and bacteria[16–18]. Based on the kinetic hypothesis and the fluxes simulated for these organisms, a fundamental model of carbon metabolism evolution for ancestral bacteria and archaea starting from LUCA is presented.

## Results

**Chemolithoautotrophic TCA fluxes are simulated with a reversible reaction of CS.** In the genomes of *T. takaii* and *D. acetivorans*, typical key enzymes involved in the rTCA cycle of autotrophs were identified, namely, the ferredoxin-dependent enzymes 2-oxoglutarate:ferredoxin oxidoreductase and pyruvate:ferredoxin oxidoreductase. On the other hand, genes for ATP citrate lyase and its two-step variant citryl-CoA synthetase/citryl-CoA lyase, which are regarded to be necessary for organisms with autotrophic rTCA cycle, were missing[13,14]: these organisms instead possessed genes of CS that were thought to be active only in oTCA cycle. In the present study, the kinetic reaction models for enzymes identified for carbon metabolism in *T. takaii* were developed (Eqs. B1–B18 in the Supplementary Information (SI)), and the kinetic network model in which these enzymatic reactions were incorporated was presented. To investigate the direction of carbon metabolic fluxes on the network, the steady-state fluxes were determined by the kinetic network model (Eqs. 1–16 in the SI) with the fixed concentrations of chemical species listed in the SI Table A1. The five universal precursors of anabolism, ACOA, pyruvate (PYR), phosphoenolpyruvate (PEP), oxaloacetate (OAA), and 2-oxoglutarate (AKG: α-ketoglutaric acid), were assumed to be consumed by biomass synthesis in the kinetic network model (Eq. B18 in the SI). The simulation utilizing the kinetic network model demonstrated the feasibility of the rTCA cycle due to a reversal of the CS reaction resulting in chemolithoautotrophic growth (Table S1a in the SI). Furthermore, the directions of obtained fluxes (Fig. 1a) were consistent with those experimentally assigned by Nunoura et al. for *T. takaii* grown chemolithoautotrophically, except for the directions between malate (MAL) and PYR and between OAA and PEP[13]. In addition, it was confirmed that the kinetic network simulations involving the gene knockouts of malic enzyme between MAL and PYR and/or of phosphoenolpyruvate carboxykinase between OAA and PEP reproduced a complete rTCA flux (Table S1b in the SI). These observations indicate that the directions of these fluxes are not essential to drive the rTCA cycle. Here the simulation with gene knockouts of enzyme means that it performed without the enzyme. Furthermore, the metabolite concentrations calculated in the kinetic network model were quantitatively consistent with those experimentally determined for *D. acetivorans* grown chemolithoautotrophically[14] (Table S2 in the SI). It was also confirmed that the autotrophic growth simulated here was robust across an extensively varied ratio of the concentrations of reduced ferredoxin ($Fdx_{red}$) and oxidized ferredoxin ($Fdx_{ox}$) (Fig. S2 in the SI). Recently, it was experimentally observed that high partial pressure from $CO_2$ drove autotrophic rTCA cycle with the reversal of the CS reaction. Our kinetic network model reproduced the $CO_2$ dependence of autotrophic growth rate observed for *D. acetivorans*[19] (Fig. S3 in the SI).

The apparent Gibbs reaction energy ($\Delta_r G_i^{tot}$; defined in Eq. A9) for the CS reaction was calculated to be −53.4 kJ/mol, and the total $\Delta_r G_i^{tot}$ of the oTCA cycle was −108.8 kJ/mol (Table S3 in the SI). The reversal of the oTCA cycle with high exergonic $\Delta_r G_i^{tot}$ requires an abundant supply of reducing agents. Reducing reactions in anaerobic organisms are more efficiently driven by reduced ferredoxins than reduced nicotinamide adenine dinucleotide (NADH) or reduced nicotinamide adenine dinucleotide phosphate (NADPH) in terms of $\Delta_r G_i^{tot}$ (Table S3 in the SI). In addition, the ferredoxin-dependent enzymes working in the rTCA cycle involve two reduced/oxidized ferredoxins in the reductive reactions (Table S3 in the SI); thus, the concentration ratio between reduced and oxidized ferredoxins needed to overcome the energetically unfavorable reaction is lower than that with NADH/NADPH.

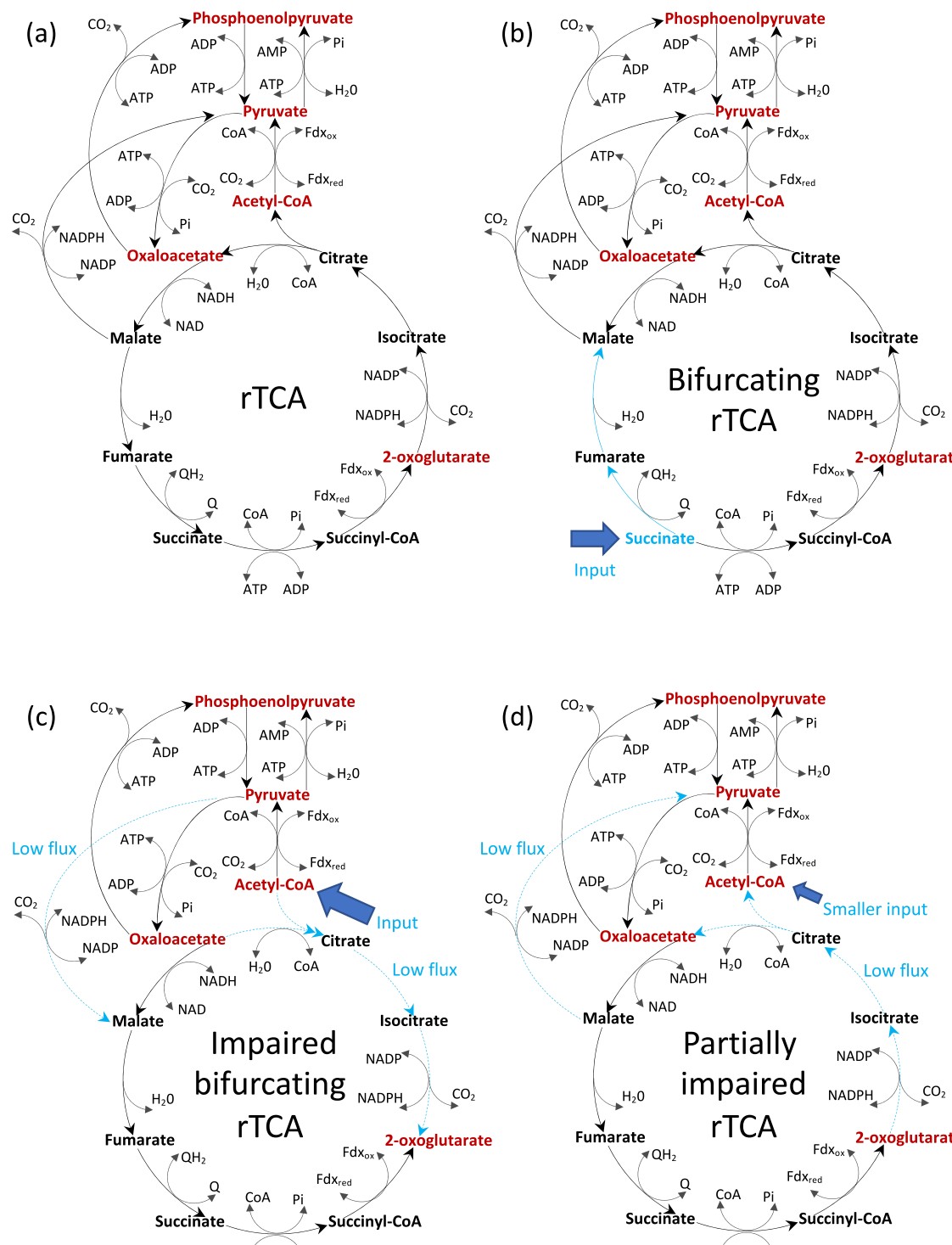

**Fig. 1 The directions of steady-state fluxes kinetically simulated for TCA cycle of _T. takaii_ and associated reactions including anaplerotic ones.**
Chemolithoautotrophic rTCA flux (**a**), chemolithomixotrophic bifurcating TCA flux caused by a succinate influx (**b**), and impaired bifurcating rTCA flux caused by an acetyl-CoA influx (**c**) and partially impaired rTCA flux caused by a smaller acetyl-CoA influx (**d**). The autotrophic growth condition same as **a** (listed by Table A1 in the SI) is used for **b**–**d**. In all the kinetic network models, the five universal precursors of anabolism, acetyl-CoA, pyruvate, phosphoenolpyruvate, oxaloacetate, and 2-oxoglutarate, are assumed to be consumed by biomass synthesis (Eq. B18 in the SI). The light blue solid/broken arrows indicate fluxes changed or significantly reduced by succinate or acetyl-CoA influx.

It is generally recognized that sufficient depletion of both ACOA and OAA is necessary to overcome the huge energetic barrier of the CS reaction and reverse it. This suggests that organisms capable of doing so must efficiently convert ACOA into PYR or cellular biomolecules and OAA into MAL[12]. To examine these hypotheses, we determined how the gene knockout of pyruvate synthase (PS), which converts ACOA to PYR, and pyruvate carboxylase (PC), which converts PYR to OAA, affects

                                              COMMUNICATIONS CHEMISTRY | https://doi.org/10.1038/s42004-021-00585-0

the rTCA cycle. It is assumed that the model of *D. acetivorans* gains a PYR influx into the rTCA cycle via serine (SER) metabolism, whereas that of *T. takaii* does not because of its lack of the necessary enzyme (Fig. S5e, f in the SI). As summarized in SI Table S4, both models with the gene knockout of PC lost the rTCA flux. However, the model of *D. acetivorans* with the gene knockout of PS maintained/reduced the rTCA flux depending on the PYR influx, whereas that of *T. takaii* significantly reduced it. These observations indicate that the depression of product concentrations in the reversed CS reaction is not sufficient to maintain the rTCA cycle and a reinvestment of the outflux equivalent of ACOA into the rTCA, which is caused by the OAA influx via PC, is additionally required. This kinetic requirement is schematically illustrated in Fig. S1a. The necessity of PC as a kinetic condition for maintaining rTCA flux revealed here plays a crucial role in interpreting the phenotype evolution of carbon metabolism for deep-branching organisms, as discussed later. However, it is not obvious whether the differences in the rTCA fluxes between *T. takaii* and *D. acetivorans* are actually due to the gene knockout of PS because the effect of PYR influx via SER metabolism on *D. acetivorans* is uncertain. Nonetheless, the investigation of the role of PC and PS by utilizing the *T. takaii*- and *D. acetivorans*-type models was useful since it shed light on the importance of PC in the rTCA cycle.

**The rTCA fluxes are bifurcated and partially impaired in the presence of succinate or acetate**. The kinetic network model with the autotrophic growth condition, as used in Fig. 1a, reproduced the directions of experimentally determined carbon fluxes for *T. takaii* grown chemolithomixotrophically in the presence of succinate (SUC) (Fig. 1b) and acetate (Fig. 1c)[13]. Upon an influx of SUC, the bifurcating oxidative flux from SUC toward MAL and the reductive flux for the remaining part of the TCA cycle were observed (Fig. 1b). The directions of the obtained fluxes were consistent with the experimental observations for *T. takaii* grown chemolithomixotrophically with SUC, except for the directions between MAL and OAA and between OAA and PYR[13]. Next, upon an influx of ACOA, the bifurcating oxidative flux from both OAA and ACOA toward AKG and the reductive flux for the remaining part of the TCA cycle were simultaneously observed (Fig. 1c). The directions of the obtained fluxes were consistent with the experimental observations for *T. takaii* grown chemolithomixotrophically with acetate, which is converted into ACOA, except for the directions between SCOA and AKG and between OAA and PEP[13]. The reductive flux was increased by the ACOA influx because of an increase in the flux through the ACOA-PYR-OAA pathway, whereas the absolute value of oxidative flux from ACOA to AKG likewise caused by the ACOA influx was $5 \times 10^7$ times smaller than that of the reductive flux, upon comparison with the reductive flux (Fig. 1c). This is because the oxidative flux by the ACOA influx was weakened by the reductive flux. Furthermore, a smaller ACOA influx did not result in a small oxidative flux but instead impaired the reductive flux from AKG toward both OAA and ACOA (Fig. 1d). This is because the smaller oxidative flux caused by the smaller ACOA influx was slightly overcome by the reductive flux. The absolute value of the impaired reductive flux was $\sim 1 \times 10^5$ times smaller than that of the reductive flux on the left side of rTCA cycle due to the competition with the oxidative flux component. This result was consistent with the experimental observation for *D. acetivorans* grown chemolithomixotrophically in the presence of acetate, where the majority of acetate uptake was mainly transformed into PYR and incorporated into the rTCA cycle via the above-mentioned pathway from PYR to OAA[14] (Fig. 1d, also see Fig. S1a in the SI). Such a conflict between the rTCA cycle and an

ACOA influx shown in Fig. 1d is held even if the ACOA influx is much smaller than the normal reductive flux from OAA to AKG (see Fig. S4 in the SI). Taken together, the ACOA influx causes a partial competition between the reductive and oxidative flux on the rTCA cycle and results in the impairment of reductive flux between AKG and OAA (ACOA). This observation further raises one kinetic hypothesis: the coexistence of either the WL or another pathway yielding ACOA influx with the rTCA cycle is kinetically inefficient because of the competition between the rTCA flux and the oxidative flux caused by the ACOA influx. Therefore, unless the ACOA influx disappears, the gene for enzymes working at the less active reaction on the rTCA cycle is lost or that of enzymes lacked on the less active part is not newly gained during evolution of LUCA. In short, the kinetic hypothesis we propose in this study is that a complete rTCA cycle is never observed in organisms because of its kinetic instability from the competition between the reductive and oxidative flux, as long as a carbon fixation pathway including the WL pathway yields ACOA influx into the rTCA cycle. The validity of this kinetic hypothesis is examined in the next subsection for deep-branching bacteria and archaea.

**TCA flux for deep-branching bacteria depends on the presence/absence of WL and rGly pathways**. To assess the validity of the kinetic hypothesis, the fluxes of carbon metabolism for deep-branching bacteria were examined using the kinetic network model simulations with enzymes identified by the KEGG genome database[15]. The results for deep-branching bacteria grown autotrophically, the Firmicutes, Cyanobacteria, Aquificae, and Proteobacteria[16,18], are shown in the SI Fig. S5. From the fluxes observed under the autotrophic growth condition (listed by Table A1 in the SI), four representative flux patterns of carbon metabolism for the deep-branching bacteria were extracted (Fig. 2). The light dashed lines without arrow in Fig. 2 and the SI Fig. S5 indicate zero flux because of either totally or partially absence of enzymes. Cyanobacteria was excluded here because it possessed an oTCA cycle[20] and was simulated with a lower reduced to oxidized agent ratio but was later incorporated in subsequent discussions. These representative flux patterns are classified according to whether the WL pathway coexists with the TCA cycle and yields an ACOA influx into the rTCA cycle. In the type-B1 pathway, the WL pathway produces ACOA influx, resulting in MAL influx through PYR into the partial rTCA pathway. This is applicable for $H_2/CO_2$-utilizing acetogens, e.g., *Sporomusa termitida*, which grows both autotrophically and heterotrophically[21]. Next, as an alternative pathway where a WL pathway coexists with CS is the type-B2 pathway, where the ACOA influx yielded by a WL pathway causes the partial oTCA pathway to synthesize AKG. However, because of the absence of several enzymes acting in the rTCA pathway, the competition between oxidative and reductive fluxes does not occur. This is found for acetogenic anaerobic bacteria, e.g., *Moorella thermoacetica*, which also grows autotrophically and heterotrophically[22]. In the type-B3 pathway, neither the incomplete WL pathway nor the incomplete rGly pathway produces an ACOA influx; hence, there is no competition between reductive and oxidative flux, and the necessary kinetic condition with respect to PC is also satisfied, thus a complete rTCA flux is produced. This is observed for sulfur-reducing bacteria, *T. takaii*[13] and *D. acetivorans*[14], which grow chemolithoautotrophically and heterotrophically. In the type-B4 pathway, if the rGly pathway produces ACOA influx, the kinetic hypothesis does not allow a complete rTCA cycle, yielding OAA influx through PEP into the partial rTCA pathway. This is observed for anaerobic anoxygenic photoheterotrophs, e.g., *Heliobacterium modesticaldum*[23,24]. Interestingly, it was

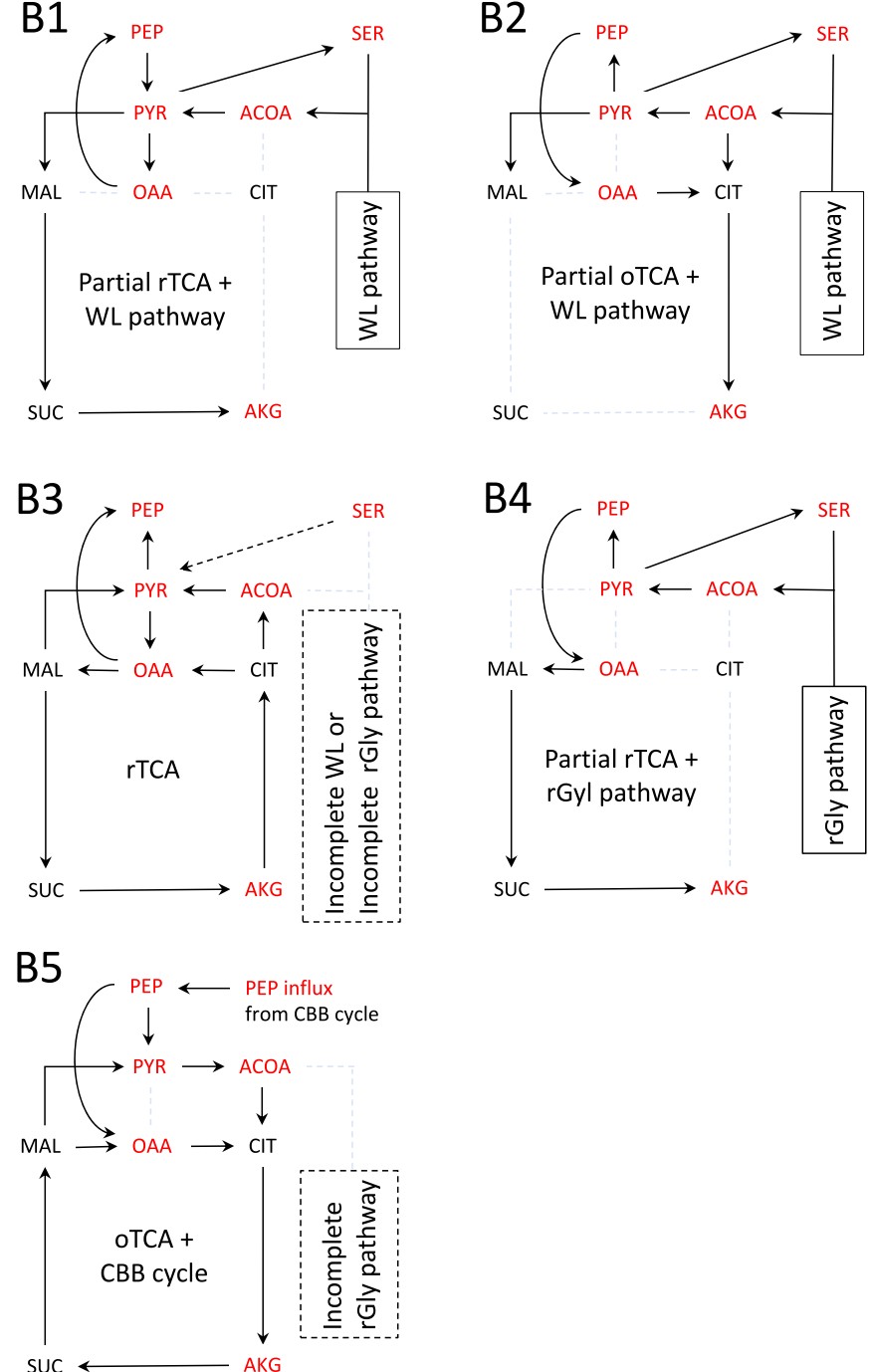

**Fig. 2 Typical flux patterns of carbon metabolism for deep-branching bacteria obtained from the kinetic network model.** B1: a partial rTCA pathway coexisting with a WL pathway that yields an ACOA influx; B2: a partial oTCA pathway coexisting with a WL pathway that yields an ACOA influx; B3: a full rTCA cycle and either an incomplete WL or an incomplete rGly pathway; B4: a partial rTCA pathway coexisting with an rGly pathway that yields an ACOA influx. The light dashed lines without arrow indicate zero flux because of either totally or partially absent enzymes. The "incomplete WL or rGly pathway" also includes cases where enzymes acting in the WL or rGly pathway partially exist, indicating no ACOA influx from outside. The dashed arrows mean that there are cases where the corresponding flux is present or absent depending on species. The autotrophic growth condition with high $Fdx_{red}/Fdx_{ox}$ ratio (listed by Table A1 in the SI) is used in the kinetic network simulations for bacteria B1–B4 grown autotrophically.

expected that, according to the KEGG genome database[15], *H. modesticaldum* possessed an rGly pathway similar to the one recently identified for *Desulfovibrio desulfuricans* grown chemolithoautotrophically[2]. The absence of complete rTCA cycle in *H. modesticaldum* possessing no WL pathway but an rGly pathway supports the extended applicability of kinetic hypothesis. Taken together, the proposed kinetic hypothesis is basically satisfied

because the network patterns observed for deep-branching bacteria do not involve impaired rTCA fluxes.

**Partial or impaired rTCA and full oTCA fluxes are observed for deep-branching archaea.** Next, the fluxes of carbon metabolism for deep-branching archaea were examined using the kinetic network model simulations with enzymes identified by the

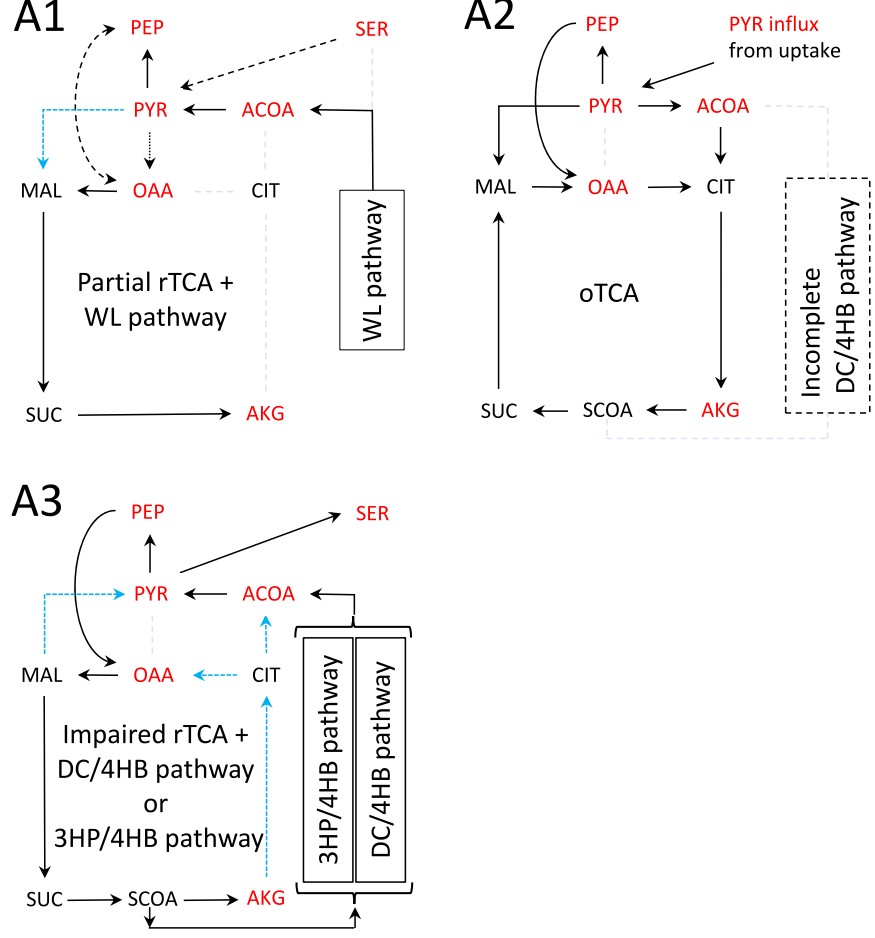

**Fig. 3 Typical flux patterns of carbon metabolism for deep-branching archaea obtained from the kinetic network model.** A1: a partial rTCA pathway coexisting with a WL pathway that yields an ACOA influx; A2: a heterotrophic oTCA cycle and an incomplete dicarboxylate/4-hydroxybutyrate (DC/4HB) pathway; A3: a partially impaired rTCA cycle coexisting with a DC/4HB or 3-hydroxypropionate/4-hydroxybutyrate (3HP/4HB) pathway. The light dashed lines without arrow indicate zero flux because of either totally or partially absence of enzymes. The black dashed arrows mean that there are cases where the corresponding flux is present or absent depending on species. The light blue dashed arrows indicate significantly small fluxes. The autotrophic growth condition with the high $Fdx_{red}/Fdx_{ox}$ ratio (listed by Table A1 in the SI) was applied for A1 and A3, while a heterotrophic growth condition with a lower $Fdx_{red}/Fdx_{ox}$ ratio was used for A2 with PYR uptake. In A3, the DC/4HB and 3HP/4HB pathways were modeled as a combination of additional ACOA influx and the conversion of succinyl-CoA (SCOA) to ACOA.

KEGG genome database[15]. The results for the deep-branching archaea grown autotrophically, namely, Euryarchaeota, Candida-tus Bathyarchaeota, and Crenarchaeota[16–18], are shown in the SI Fig. S6, except for Fig. S6f. Based on the fluxes observed under the autotrophic growth condition (listed by Table A1 in the SI), three representative flux patterns of carbon metabolism for deep-branching archaea were identified (Fig. 3). The light dashed lines without arrow in Fig. 3 and SI Fig. S6 indicate zero flux because of either total or partial absence of enzymes. In the type-A1 pathway, the WL pathway produces an ACOA influx, resulting in an OAA influx into the partial rTCA pathway. This is found for auto-trophic methanogens such as *Methanococcus maripaludis*[25–27]. In the type-A2 pathway, the absence of WL pathway does not pre-vent a complete rTCA cycle under autotrophic growth conditions. Nevertheless, the rTCA cycle is kinetically inhibited because of the absence of PC in this pathway. This absence led to the loss of the OAA influx from PYR into the rTCA cycle, thus causing a decrease in the concentration of CIT, which is the product of rTCA cycle, and resulted in the impairment of the reversed CS reaction. Consequently, the right half of the rTCA flux was sig-nificantly reduced (also discussed in the SI Table S4). In fact, as shown in the type-A2 pathway, instead of the rTCA cycle, the

complete oTCA cycle under a heterotrophic growth condition was reproduced even in the absence of PC by an uptake of PYR and lowering the ratio of reduced agent ($Fdx_{red}$) to oxidized agent ($Fdx_{ox}$). This is found for the heterotrophic anaerobic archaeon *Vulcanisaeta distributa*[28].

As shown in the type-A3 pathway, the partially impaired rTCA flux was observed under the autotrophic growth condition even in the absence of the WL pathway. In addition to the absence of PC, type-A3 Crenarchaeota *Thermoproteus tenax* and *Acidianus hospi-talis* possess the dicarboxylate/4-hydroxybutyrate (DC/4HB)[29] and 3-hydroxypropionate/4-hydroxybutyrate (3HP/4HB) pathways[30], respectively. These alternative carbon fixation pathways yield an ACOA influx under chemolithoautotrophic growth conditions and thus impairs the rTCA flux on the right-hand side (Fig. 3 (A3)) in the same manner as the WL pathway. The inefficiency on the partially impaired rTCA cycle should be acceptable if the DC/4HB and 3HP/4HB pathways more efficiently work for carbon fixation. Further-more, these Crenarchaeota grow heterotrophically[29,30]. The oTCA flux was simulated under heterotrophic growth conditions with lowering the $Fdx_{red}/Fdx_{ox}$ ratio, though not shown in Fig. 3 (A3). Therefore, it is conceivable that these Crenarchaeota possess and maintain the enzymes for the less active reductive reactions on the

partially impaired rTCA cycle to utilize the oTCA cycle under heterotrophic growth conditions. Taken together, the kinetic hypothesis is satisfied on the carbon metabolic pathway of the deep-branching archaea, parallel to that of bacteria.

## Discussion

**Elementary evolution processes of carbon metabolism are derived from the kinetic hypothesis and the simulated typical flux patterns.** In the pioneering work by Braakman and Smith[5], a single connected redundant network consisting of a complete WL pathway and a full rTCA cycle was proposed as a more robust and plausible topology that LUCA or (proto)biological systems leading up to the LUCA should possess on the basis of phylometabolic analysis. In contrast, the results obtained from our kinetic network model do not support the coexistence of those pathway in one organism, unless a more dominant and selective pressure that favors the combination of those pathways exists in parallel. The key difference between their work and ours is whether the kinetic factors are taken into consideration. If it is assumed that, as widely accepted, LUCA is an anaerobic hydrothermal chemoautotroph possessing at least a WL pathway[1], the confirmed kinetic hypothesis raises a model of evolution processes on carbon metabolism for deep-branching archaea and bacteria (Fig. 4). In this model, the simulated typical carbon metabolism pathways A1–A3 (Fig. 3) B1–B4 (Fig. 2), and type-B5 were incorporated. The type-B5 pathway is observed for Cyanobacteria, e.g., *Synechocystis* sp., which possesses an oTCA cycle and a Calvin–Benson–Bassham (CBB) cycle (Fig. S5d in the SI)[20].

As shown in Fig. 4, the evolution processes observed for deep-branching archaea and bacteria can be classified into two types: type-I evolution that occurs with a coexisting complete WL pathway and type-II evolution that occurs in the absence of a complete WL pathway. The incomplete WL (iWL) pathway often lacks, at the very least, CO dehydrogenase/acetyl-CoA synthase, which is the final step of ACOA synthesis and is known as one of the most oxygen-sensitive enzymes in the biosphere[5,31]. Braakman and Smith proposed that oxygen toxicity to the enzymes act as a selection force on the iWL pathway[5]. First, we discuss how type-I evolution occurs in the presence of a WL pathway from the viewpoint of the kinetic hypothesis. The phenotype that possesses a partial rTCA pathway coexisting with a WL pathway is commonly observed for both archaea and bacteria (A1 and B1) and is thus predicted to be a plausible candidate for the carbon metabolism phenotype of LUCA. Alternatively, if LUCA does not possess a partial rTCA pathway, such ancestral chemoautotrophs would gain enzymes necessary for a partial rTCA pathway that coexists with the WL pathway during evolution toward A1 or B1. On the other hand, in cases where ancestral chemoautotrophs gained CS during the early stages of type-I evolution, they would gain enzymes necessary for a partial oTCA pathway to synthesize AKG, as seen in B2. This corresponds to acetogenic anaerobic bacteria growing autotrophically and heterotrophically (Fig. S5d in the SI).

**Evolutionary divergence of ancient carbon metabolism not coexisting with WL pathway.** Here it should be noted that the kinetic hypothesis does not exclude type-II evolution from LUCA, which is a direct evolution from an ancestral chemoautotroph coexisting with an incomplete WL pathway. Therefore, type-II evolution occurs following type-I or could take place directly from LUCA as well. As seen in Fig. 4, type-II evolution of bacteria depended on whether they obtained PC, which was necessary for maintaining a full rTCA cycle, if no ACOA influx was supplied by the other carbon fixation pathway. If ancestral bacteria could gain PC or keep that was previously gained, they would gain enzymes

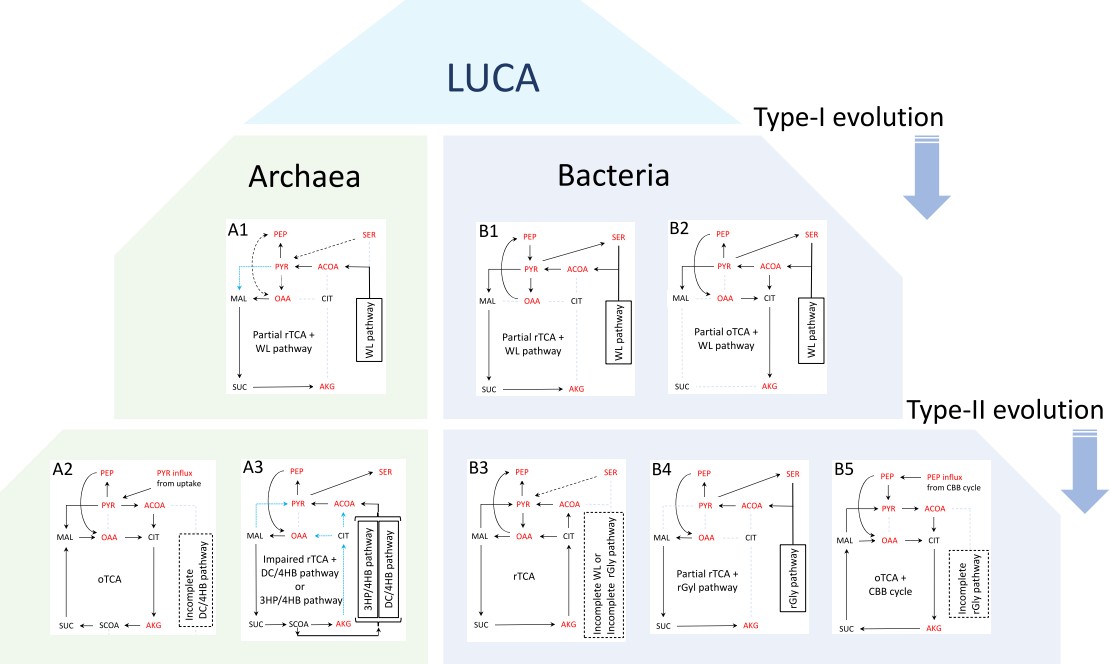

**Fig. 4 Evolution process model from LUCA on carbon metabolism pathway for deep-branching archaea and bacteria.** In this model, it is assumed that LUCA is an anaerobic hydrothermal chemoautotroph that possesses a WL pathway[1]. Evolutionary processes observed for deep-branching archaea and bacteria are classified into type-I and type-II evolution: the former and latter occur in the presence and absence of a WL pathway, respectively. During type-I evolution, the WL pathway yields direct ACOA influx into the rTCA cycle, while it does not do so in type-II. Type-II evolution occurs following type-I evolution or could take place directly from LUCA as well. The phenotypes B1–B4 and A1–A3 are the same as shown in Figs. 2 and 3, respectively.

necessary for a full rTCA cycle to increase the rTCA flux (phenotype B3); otherwise, they would undergo alternative evolution. On one hand, as seen in B5, because the ancestor of Cyanobacteria without PC acquired a CBB cycle that enabled of photoautotrophic growth, utilizing neither WL nor the rTCA carbon fixation pathway, they would gain enzymes for an oTCA cycle to produce energy, reducing agents, and metabolites so that they grew photomixotrophically as well (Fig. S5d in the SI)[32]. On the other hand, as seen in B4, some ancestral bacteria acquired an rGly pathway[2], which worked as an alternative carbon fixation pathway and provided an ACOA influx, and they either gained enzymes necessary for partial rTCA or kept the ones already possessed so that they synthesized AKG. *H. modesticaldum*[24] with the type-B4 pathway possesses enzymes for the remaining part of the rTCA cycle, except for CS and PC. Thus, *H. modesticaldum* might have evolved from type-B3 pathway and, during this process, possibly lost PC and a part of the less active enzymes of the impaired rTCA cycle.

Turning to the phenotype of deep-branching archaea, those with type-A3 have rTCA flux partially decreased by both the ACOA influx from a DC/4HB or 3HP/4HB pathway and the lack of PC. Therefore, from a kinetic point of view, it would not be expected that they directly gained the enzymes with significantly low activity during evolution. Type-II evolution of ancestral archaea is thus interpreted according to the following plausible scenario: as seen in A2, ancestral archaea without PC would first gain enzymes necessary for a complete oTCA cycle under heterotrophic growing conditions so that they evolved to be heterotrophic anaerobic archaea. Subsequently, the heterotrophic anaerobic archaea would acquire enzymes necessary for a DC/4HB or 3HP/4HB carbon fixation pathway and gain the ability to grow not only heterotrophically but also chemolithoautotrophically, as seen in A3. Notably, *V. distributa*[28] with type-A2 partly possesses enzymes necessary for the DC/4HB carbon fixation pathway. Thus, it might have evolved into a heterotroph from type-A3 and, during the process, possibly lost some of the less active enzymes for the DC/4HP pathway under heterotrophically growing conditions.

In summary, the model of elementary evolutionary processes on carbon metabolism presented here satisfies the following two kinetic conditions that are necessary for maintaining a full rTCA cycle: avoiding a direct ACOA influx into the rTCA cycle and the reinvestment of the outflux equivalent of ACOA into the rTCA via the OAA influx. These kinetic hypotheses and necessary conditions are expected to play crucial roles in revealing the carbon metabolic network that LUCA possesses and understanding the evolution processes from LUCA toward deep-branching archaea and bacteria. However, the selective forces such as energy optimization, oxygen toxicity, and the alkalinity in environments[5], and influences of lateral gene transfer[33] are not taken into consideration in the evolution processes presented in Fig. 4. Investigation of the relation among the presented kinetic insights, the abovementioned selective forces, and evolutionary convergences are the scope of future research.

## Methods

**Kinetic network model.** The enzymatic reactions identified for the rTCA cycle of *T. takaii* and the reactions related to the TCA cycle including anaplerotic ones (Eqs. B1–B17 in the SI) were incorporated into the kinetic network model for carbon metabolism described by the ordinary differential Eqs. 1–16 listed in the SI (also see Fig. A1 in the SI). The information on the enzyme-coding genes were obtained by browsing on specific organism pages in the KEGG database[15]. In addition, biomass synthesis reactions for cell growth (Eq. B18 in the SI) were also incorporated into the kinetic network model (see Fig. A1 in the SI). Thus, the five universal precursors of anabolism, ACOA, PYR, PEP, OAA, and AKG, are consumed by the biomass synthesis. Kinetic models for enzymatic reactions on the rTCA cycle were, in part, taken from the literatures including the modeling paper of Beard and co-workers[34], and those that were not available from literatures were developed by ourselves on the basis of biochemical experimental data. The details

of model equations and parameters used in the simulations are given below. The key point that should be especially stressed on the kinetic network modeling was the apparent equilibrium constants upon total concentrations of substrates and products for each enzymatic reaction, which were precisely determined according to the method and database developed by Beard and co-workers[35]. The calculation of the apparent equilibrium constant and the apparent reaction Gibbs free energy is explained in "General aspects on kinetic modeling of enzymatic reactions" of the SI.

**Simulations.** The ordinary differential equations (Eqs. 1–16 in the SI) for the kinetic network model comprised of 22 reaction equations and 29 chemical species were solved using COPASI biochemical system simulator (ver. 4.28)[36]. The steady-state concentrations and fluxes for all the organisms of which results were discussed in the main text and SI were determined by the kinetic network model simulation based on Eqs. 1–16 in the SI. The concentrations of chemical species used as the model parameters in the kinetic network model are listed by Table A1 in the SI. Several model parameters in Table A1 were estimated so that the kinetic network model simulation reproduced the concentration of metabolites experimentally determined for *D. acetivorans* grown autotrophically (Table S2 in the SI)[14].

## Data availability
The input data for COPASI used to generate the data in the current study are available from the corresponding author, TS, upon reasonable request.

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

## Acknowledgements

This work was supported in part by the JSPS KAKENHI Grant No. JP20K05431.

## Author contributions

T.S. developed the kinetic network models, performed the simulations, and analyzed the computational data. T.S. and K.H. discussed the results and wrote the paper.

## Competing interests

The authors declare no competing interests.
