## [Peer Review File · Communications Chemistry]

Reviewers' comments:

Reviewer #1 (Remarks to the Author):

Summary of Article: In this manuscript, the authors reconstructed a kinetic model to assess the importance of decoupled reductive tricarboxylic acid cycle (rTCA) and the Wood–Ljungdahl pathway (WL) in the evolutionary history of ancient chemolithotrophic and chemoautotrophic bacteria and archaea.

Summary of reviewer Comments: This article is generally well written and explores the interesting topic of the Last Universal Common Ancestor (LUCA), which could make this of interest to a great number of readers. The biggest concern regarding this manuscript is the absence of a complete description of the methodology. The methods section is severely inadequate in explaining how the kinetic model was developed, how and which parameterization processes were involved, or the details on how the system of differential equations were solved. The section needs significant overhaul to include all these details.

Major Critiques:

- 1) The authors also need to explain their method of estimating the free energy change values. What algorithm was used? What substrate concentrations or other physiological conditions were used?
- 2) Source 1 hypothesizes the genome of LUCA, as well as presents a hypothesis of its metabolism reconstructed from genomic data. If the goal is to highlight how the evolution of carbon metabolism occurs from LUCA, some schematic showing the different evolutionary pathways from LUCA's metabolism to that of modern species would be extremely helpful and instructive for readers.

Minor Critiques:

- 1) In the results section, the following statement seems to be poorly explained: "The reductive flux was increased by the ACOA influx because of the resulting OAA influx, whereas the oxidative flux was obviously suppressed by the conflict with the reductive flux." Why did the authors conclude that the increase in the reductive flux and suppression of oxidative flux is a result of the OAA influx? Is this a potential mechanism that the authors are proposing or is there any evidence that suggests this mechanism?
- 2) A direct statement of "the kinetic hypothesis" would go a long way toward attempting to clarify what this work seeks to achieve, e.g. "the kinetic hypothesis we investigate in this study is ...".

Reviewer #2 (Remarks to the Author):

This paper examines how changes in the in- and outflow of various intermediates of the citric acid (TCA) cycle affects its dynamics, and considers implication thereof for the early evolution of metabolism. As I understood it, a kinetic model was built for a network of the TCA cycle and a few related reactions (the network contains 22 total reactions), and its dynamics were evaluated under a number of perturbations, including "knocking out" reactions, or changing influx of some metabolites. In particular the authors examined under which conditions flux through the reductive TCA (rTCA) cycle was possible. They conclude that flux through the rTCA cycle is not possible under excessive influx of acetyl-CoA, which they in turn conclude presents a kinetic barrier to the co-

occurrence of the rTCA cycle and the reductive acetyl-CoA (“Wood-Ljungdahl”) pathway in the same organism. The authors then perform a genomic survey of bacteria and archaea, which they conclude is consistent with the modeling results. Based on these conclusions the authors end by presenting some hypotheses regarding the early evolution of CO₂-fixing pathways.

The ideas are interesting, and some of the conclusions seem plausible in principle (others might be debated), but I must admit that in practice I found the paper difficult to evaluate. This is mainly because I often found it hard to follow what was done and how the authors arrived at their conclusions. Technical details and motivations for the approaches are somewhat lacking, and figures only really show schematics and diagrams. The text also typically jumps quickly to the conclusion, without guiding the reader through the approach and key assumptions, as well as the path from results to conclusions. While the paper is a metabolic modeling paper and much of the discussion focuses on fluxes that are derived, I cannot find results for calculated fluxes under the various changes imposed on the network anywhere in the paper. Even the supplement only shows schematic figures of “typical flux patterns”. To be completely honest, I am somewhat left wondering what the actual results were...

In short, I think the ideas in this paper could ultimately make for a nice contribution to the broader discussion on the early evolution of metabolism, but as it stands the ideas are a little hard to evaluate. I would recommend the authors provide greater clarity throughout on the approach and assumptions, show more direct results and provide further discussion thereof, and generally more carefully guide the reader through the logic, before diving into conclusions and hypotheses. Some specific comments are given below. All are offered in the spirit of trying to help the authors improve their manuscript.

All the best,
Rogier Braakman

General comments:

1. I did not fully understand how the modeling approach works. A typical example of the general issues highlighted above comes from the following sentences in the first paragraph of the Results section “In the present study, a kinetic network model with enzymes identified in *T. takaii* was developed, and the feasibility of the chemolithoautotrophic rTCA cycle due to a reversal of the CS reaction was demonstrated (Fig. 1a). The obtained fluxes were consistent with the direction experimentally assigned by Nunoura et al., except for the fluxes between malate (MAL) and pyruvate (PYR) and between oxaloacetate (OAA) and PYR.” So, over the course of just two sentences the text goes almost directly from mentioning what was done to the results, but for me it left many questions about the method unanswered:

- What exactly do you mean by kinetic model in this case? It would really help the reader if you could explain at a high level how this approach works. Indeed, given that this modeling effort forms the heart of the whole paper, spending a full paragraph just explaining how it works would not be inappropriate. Even after reading the Methods section and Supplement it was still not entirely clear to me. I finally understood you built a model of 22 total reactions (when reading the main text, I first thought you built full genome-scale models), including a “biomass synthesis” reaction. Does the latter imply you took an FBA-type approach to the modeling?

- I also think I understood that the model was built on a foundation in which the free energies (and thus directionality) of reactions were treated as variables that depend on the concentrations of reactant, but it wasn't clear to me how you then explored the model dynamics. That is, a single list of free energies of reaction was given in the supplement, but of course reaction free energies change depending on the concentrations of the reactants, which is part of what you are modeling, so how did the modeling work in practice? Did you take some kind of recursive modeling approach until the network equilibrates? And given that several simulations were done, with each solution presumably having their own Gibbs free energies of reaction, why were only single values listed?

- Related to the previous question, I was generally a bit confused about how Gibbs free energies were used, since the text talks about "standard Gibbs free energies" (ΔG^0) which are for standard conditions (importantly including reactant concentrations of 1M, far above the more typical mM levels of cells), and the legend of Table S2 similarly seems to suggest the listed values are standard Gibbs free energies of reaction $\Delta_r G^0$. However, equations A7-A9 suggest you are calculating the Gibbs free energies of reaction using physiologically relevant concentrations. But then this comes back to my previous point that each network solution presumably has its own set of Gibbs free energies, since concentrations are changing, making me wonder why only a single list of values is given in Table S2. It would be good if you make sure this is really clear throughout.

- Stepping back, did you solve for the basic ability to grow (i.e. can biomass be produced yes/no) and then examine fluxes that emerge in the model solution, or did you systematically vary metabolite concentrations to see which combinations allowed growth? Relatedly it was also not clear to me which metabolite concentrations you fixed and which you allowed to vary, and for those that you allowed to vary how their ranges were constrained to maintain physiological realism.

Anyway, my larger point is that such details should be really clear to the reader so they can evaluate the approach and thus the results/conclusions coming from it. As you can tell it took me quite some effort to get my understanding to where it is, and I'm still somewhat confused... There are other examples like the sentences highlighted at the start of this point that I tried to further highlight in the specific comments below.

2. The kinetic hypothesis states that because influx of acetyl-CoA acts to inhibit reductive flux through the TCA cycle, the linkage between rTCA and WL is disfavored. But, organisms with the WL pathway generally redirect acetyl-CoA/acetate flux toward other endpoints that are released from the cell, e.g. acetate \rightarrow methane (methanogens) or acetyl-CoA \rightarrow acetate \rightarrow excretion (acetogens), which indicates that acetyl-CoA might accumulate much less in those cases than is suggested in the text here. Did you explore this possibility? I couldn't tell if the kinetic model here has the possibility of such excretion mechanisms? If you added them, would it affect your results and therefore your conclusions/hypothesis?

- Relatedly, the text states that simulated concentrations of key metabolites are consistent with concentrations measured experimentally. However, looking at Table S1, while concentrations of citrate, succinate and malate are indeed relatively similar between model and experiment, the concentration of acetyl-CoA is nearly an order of magnitude higher in the model than in the experiment, and it is acetyl-CoA that is the key intermediate in your argument for a kinetic conflict between the rTCA cycle and WL pathway. This discrepancy and how it affects simulations under

enhanced acetate/succinate influx might deserve further comment. Are you sure it does not affect the results?

3. I also did not fully understand what methods were used in surveying the combination of WL and TCA variants used in extant bacteria and archaea (i.e. pages 6-9). The first sentence of this section states "To assess the validity of the kinetic hypothesis, the fluxes of carbon metabolism for deep-branching bacteria according to the enzymes identified through the KEGG genome database were examined", and the following sentence begins discussing results. But, as far as I can tell, the method is not actually described anywhere, including in the methods section. I am again left with several questions:

- did you perform kinetic modeling runs for these networks as in the previous section, or did you do metabolic network reconstructions from genomes, or both? The discussion of "obtained fluxes" throughout this section implies numerical modeling was done, while the use of KEGG implies network reconstructions. If the work done here was solely network reconstruction, it would be best if the text avoids use of "fluxes" and "flux patterns" to describe the results, since those suggest modeling results. This phrasing in terms of fluxes is also used in the previous section where modeling was done, and both sections show schematic flux diagrams as the main results, which is partly where the confusion comes from. If modeling and calculations were in fact done (or also done, alongside network reconstructions), then the results should be listed somewhere beyond only the schematic diagrams. The supplement currently only contains one network variant (that from the first section), so if other variants were also modeled those results should also be shown somewhere.

- If the work indeed consisted of metabolic network reconstructions, more details on the approach should be given, at the very least in the Methods section. Was it done via web-browsing on the various organism pages in the KEGG database? Or did you perform BLAST searches of reference genes/enzymes in the given organisms? It should be noted that reconstructions based on web-browsing of KEGG has some uncertainties due to misannotation of genes. It is therefore generally advisable to follow up web-browsing to get first impression with more careful sequence-based searches and analyses. I cannot quite tell which approach was used in this case.

4. On a more conceptual side, I also didn't fully understand aspects of the discussion around evolution of pathway variants. The paper concludes there is a kinetic conflict between the rTCA cycle and WL pathways that prevents both from co-occurring in the same organism, which is an interesting conclusion if it holds up around clarifications mentioned above. However, much of discussion in the final two parts of the result section (i.e. the KEGG-based surveys of bacteria/archaea) and the discussion section focuses on the "avoidance" of this conflict over the course of evolution. I'm not sure I understood this point, as it implies that there is in fact some other selective pressure acting in parallel that favors the combination of the rTCA cycle and WL pathway. If there isn't some intrinsic benefit to the direct linkage of the full rTCA cycle and WL pathway, why and how would a conflict ever arise? Indeed, use of the term "conflict" implies there are two forces working opposition to each other, one in favor of linking the pathways, the other in favor of breaking that linkage. The only other scenario in which I can see "evolutionary avoidance" of the kinetic conflict coming into play is if at some point in early evolution the two pathways did co-occur in the same cell, in which case evolution away from that state (at least partly driven by the need to remove the previous conflict) could have created all the disconnected variants seen today.

- Directly related to the previous point, the text states in the first paragraph of the Discussion section that the results obtained here are inconsistent with the conclusions in Braakman and Smith 2012, which I also didn't fully understand. I generally don't bring my own research directly into the review process, but since the text directly addresses our paper and I happen to be reviewing this paper, I will briefly respond. In discussions on the early evolution of metabolism it is important to distinguish between two major phases of evolution: 1) pre-LUCA evolution, which reaches back to the origin of life and proceeds via pre- and proto-cellular stages of chemical self-organization to the emergence of fully integrated cells, and 2) post-LUCA evolution, which involves evolutionary diversification away from the LUCA after cellular life was robustly established. Like the current paper, our 2012 paper reconstructs evolution of CO₂-fixing pathways back from extant life to the LUCA, in that case identifying an integrated rTCA+WL pathway as the most likely ancestral CO₂-fixing pathway. However, the subsequent statements about the "more robust and plausible network topology" of this network refer to pre-LUCA evolution. That is, we argued that the rTCA+WL network would have benefited from a built-in network-topological robustness during the process of chemical self-organization leading from prebiotic chemistry to fully integrated cellular life. Once fully integrated cells had emerged, that network-topological robustness would have become redundant, and, indeed, post-LUCA evolution produced a number of variants in which the rTCA and WL pathways have become disconnected from one another. The current paper similarly examines post-LUCA evolution, since it surveys diversity in extant metabolisms, so it's not totally clear to me why the results are inconsistent with what we found in our paper. Indeed, the papers actually seem highly compatible, with the main difference lying in the emphasis on different (but again compatible) forces driving the disconnection between the rTCA cycle and WL pathways. Anyway, this is not meant to push the views of our paper – the authors should obviously feel free to discuss their results and speculate as they see fit, and indeed it is good for discussion, debate and disagreement to continue in the larger community - but i did want to highlight how it is important to distinguish between pre- and post-LUCA evolutionary processes when discussing most likely properties of the LUCA. LUCA was by definition the product of pre-LUCA evolution, and so if the current paper wants to draw conclusions about what was the most likely pathway in the LUCA, it would be good to explicitly consider how and why forces found to have shaped post-LUCA evolutionary diversity should be extended back to understanding evolutionary processes in pre-LUCA times.

5. A somewhat smaller point, but i just wanted to note that in the Discussion section there is discussion of evolutionary processes in terms of various lineages "trying to acquire" or "securing" various enzymes. This phrasing endows microbes with a lot of agency and foresight, and gives the impression that cells are able to "choose" their evolutionary trajectories. I'm not sure this was intentional? It might be good to double check the language used to discuss evolutionary dynamics.

Remaining specific comments:

- first paragraph of results, middle of p. 3: What do you mean by "feasibility was demonstrated"? The text only cites Fig. 1A for this claim, but it shows only a schematic diagram. Where are the actual results shown? What were your key assumptions and how did you test if the citrate synthase reaction could be reversed? Did you vary concentrations, or something else? This should all be explained more clearly, see previous comments

- also first paragraph of results, middle of p. 3: What do you mean by the "fluxes were consistent"

with previous experimental results? Do you mean just direction, or also magnitude? Please clarify in the text.

- also first paragraph of results, middle of p. 3: The text contains the statement that the direction of two fluxes in the TCA cycle (between MAL and PYR and between OAA and PYR) found in the simulations are not consistent with the experimental findings of Nunuora et al. Does this not invalidate the conclusion? I don't understand how full rTCA cycling can be maintained if the flux of two reactions is reversed. It would be great if this could be addressed a bit more.

- bottom of p. 3/top of p.4: the text states that the rTCA flux was evaluated under various gene knockouts. How was this done? I cannot find any details. The text also states that results are listed in Table S3, but table S3 only contains text entries that flux was impaired/reduced/lost. Actual results again seem to be missing.

- section on TCA cycle bifurcation, middle p. 4: the text states that the kinetic model reproduced results in presence of succinate and acetate, but again only points to a schematic diagram. Where are actual results shown? And again, do you mean direction, or magnitude of fluxes?

- section on TCA cycle bifurcation, middle p. 4: can you elaborate a bit on the underlying molecular mechanisms that drive the bifurcations? Do concentrations of some intermediates increase more than others, thereby causing the flux to reverse? In principle one could imagine influx of acetate/succinate simply driving the rTCA cycle forward, so it would be great to have a bit further discussion, since modeling studies like in this paper seem ideally suited to disentangle the subtleties of the dynamics.

- section on TCA cycle bifurcation, middle p. 4: Like in the previous section, the text here states that the direction of two fluxes in the TCA cycle (between MAL and PYR and between OAA and PYR) found in the simulations are not consistent with the experimental findings in *T. takaii*. Does this again not invalidate the conclusion? See previous comment...

- section on surveys in other bacteria/archaea, top of p. 6: why were cyanobacteria first excluded and then reincorporated into the analysis? This was not clear from the text

Reviewer #3 (Remarks to the Author):

Recently, it was shown that the oxidative, citrate synthase-dependent TCA cycle can be reversed to fix inorganic carbon, allowing autotrophic growth. This finding implies a lot of kinetic and thermodynamic issues that need to be addressed in modelling and kinetic studies. Therefore, the manuscript "Kinetic study on ancestral carbon metabolism pathways in deep-branching bacteria and archaea" is an important and timely study. I like the idea of the manuscript and support its publication. However, I had a hard time to understand what the authors want to say, and in many cases I am not sure whether I have really got what they meant. I am not an expert in kinetics studies, modelling and related topics but I think that I am not a layman in microbial metabolism. Therefore, I believe that the manuscript needs many more explanations that will help the reader to follow it. Also, it should be better structured and be carefully proofread, as it is now full of various mistakes. Starting with the claim that the rTCA cycle and the WL pathway belong to the most ancient metabolic pathways, the authors analyze a possible co-occurrence of these two pathways in LUCA,

and this causes a question how the authors actually came to the idea to analyze it. There is no single organism that has both pathways at once, and the reason for having them in one cell is at best not obvious. In fact, the co-existence of two pathways in one organism was not unequivocally demonstrated in any organism in general, although there are good reasons to believe that this is the case for the Calvin cycle and the rTCA cycle in some sulfur bacteria. The proposal of Braakman and Smith of the co-functioning of WL and rTCA in LUCA that justifies analysis is mentioned first in the discussion. I think that it has to be mentioned and briefly discussed in the Introduction, in order to avoid the misunderstanding.

p. 2: „Currently, six different autotrophic pathways responsible for carbon fixation are recognized“: with the glycine reductase pathway, seven (DOI: 10.1038/s41467-020-18906-7).

p. 2: “In the present study, kinetic simulations of the autotrophic rTCA cycle for these bacteria are presented, and the mechanism by which the CS reaction is reversed, which was thought to function only in the oxidative direction, is revealed in dynamical systems”: you may wish to discuss the recent paper of Steffens et al showing that high CO₂ partial pressure allows CS to be reversed (doi: 10.1038/s41586-021-03456-9).

p. 2: „direct connection of the WL pathway to the rTCA cycle”: I am not sure what is meant here. What is this “direct connection”, taking into account that the WL pathway and the rTCA cycle both have the same product (acetyl-CoA)?

p. 3: “In the present study, a kinetic network model with enzymes identified in *T. takaii* was developed, and the feasibility of the chemolithoautotrophic rTCA cycle due to a reversal of the CS reaction was demonstrated (Fig. 1a)”: I am not sure how this model works. What comes in and what comes out? What is included in the model? What are the premises and assumptions? Is autotrophic growth meant here? Please explain it in details.

p. 3: “In addition, the ferredoxin-dependent enzymes working in the rTCA cycle involve two reduced/oxidative ferredoxins in the reductive reactions (Table S2 in the SI); thus, the concentration ratio between reduced and oxidative ferredoxins needed to overcome the energetically unfavorable reaction is lower than that with NADH/NADPH”: what is “oxidative ferredoxin”? Table A1 shows certain values for concentrations of reduced and oxidized ferredoxin. Where do these values come from? It’s claimed that they are “estimated for rTCA cycle of *T. takaii*”. Where and how were they estimated? The given ratio Fd_{red}/Fd_{ox} is 50,000, basically meaning that ferredoxin is reduced to 100,00% in the cell. How could it be achieved?

p. 3: “It is noted that *D. acetivorans* possesses a serine synthesis pathway that results in a PYR influx into the rTCA cycle, whereas *T. takaii* does not (Figs. S2e and S2f in the SI)”: what is “serine pathway”? The mechanisms that (in my imagination) can allow “pyruvate influx” in an autotrophic bacterium in the absence of pyruvate synthase are anaplerotic reactions functioning during growth on acetate (the glyoxylate cycle, the ethylmalonyl-CoA pathway). However, I believe that it is not what is meant here?..

p. 4: “In contrast, the model of *D. acetivorans* with the gene knockout of pyruvate synthase maintained the rTCA flux, whereas there was a decrease with *T. takaii*”: how is it possible? What is the difference between *T. takaii* and *D. acetivorans* in this regard? In fact, the mentioned Steffens et al paper highlights the essentiality of the pyruvate synthase reaction in *D. acetivorans*.

In the manuscript, the importance of pyruvate carboxylase is discussed. However, another widespread enzyme catalyzing C₃ conversion to C₄, PEP carboxylase, is not even mentioned. PEP carboxylase is probably not the best choice for bacteria using citrate synthase version of the rTCA cycle, but this enzyme does function in many organisms for autotrophic CO₂ fixation. In addition, malic enzyme can actually also work backwards, being an attractive alternative to the PC reaction.

p. 4: “The obtained results were consistent with the experimental observations for *T. takaii* grown chemolithomixotrophically with SUC, except for the fluxes between MAL and OAA and between OAA

and PYR”: could you please explain what does consistent exactly mean in this case, and what is wrong with fluxes between malate, oxaloacetate and pyruvate. The same for acetate.

In Fig. 1b, CS is shown to function backwards during growth on succinate in the figure. What are evidences for it?

Figs. 1c, 1d: the flux only in the direction of acetyl-CoA carboxylation is shown. However, if I understand the Nunuora et al (2018) data correctly, the presence/dominance of 4x13C in Glu after growth on fully labelled acetate suggest the functioning of CS in oxidative direction under these conditions. Also, the CS activity was very high on acetate, and the functioning of oxidative TCA cycle appears to be unavoidable. Doesn't the model contradict to the experimental data?

p. 4: “the ACOA influx that was smaller than that used in Fig. 1c”: could you please define this “smaller”?

p.4: “This result was consistent with the experimental observation for *D. acetivorans* grown chemolithomixotrophically in the presence of acetate, where the majority of acetate uptake was mainly transformed into PYR and incorporated into the rTCA cycle via the abovementioned pathway from PYR to OAA (see Fig. S1a in the SI)”: if I understand it correctly, Fig. S1a shows autotrophic growth of *D. acetivorans*. Acetate growth is shown in Fig. S1b, and it proceeds via the oxidative, and not reductive TCA cycle. In fact, CS would not work in the direction of citrate cleavage in the presence of acetate/acetyl-CoA. This is simply the results of CS equilibrium, and additional “kinetic hypothesis” appears to be redundant here.

p. 4: “direct connection is disconnected”: ?

p. 6: “The results for bacteria belonging to Firmicutes, Cyanobacteria, Aquificae, and Proteobacteria, which are expected to be close to LUCA (Ref. 15,17)”: in fact, the references 15 and 17 does not discuss LUCA at all. I am also wondering how the proposed close connection of e.g. Cyanobacteria to LUCA is compatible with LUCA having the WL pathway and the rTCA cycle. The same is true for p. 7, concerning Archaea.

p. 6: “whether the WL pathway connects the TCA cycle, resulting in a direct ACOA influx”: I am not sure what is meant here.

p. 6: incomplete WL pathway: what do you mean? Some of the enzymes of the pathway are involved in C1 metabolism in many (most?) organisms; however, this is not called as the WL pathway. The synthesis of serine (that is shown in Fig. 2 for this “iWL pathway”) requires glycine. Where does it come from? Or, do you mean the glycine synthesis via the glycine cleavage system? The corresponding pathway leading to serine was indeed discussed on several occasion and reconstructed in several metabolic engineering studies but has never been not shown (until now?) for natural CO₂ fixation. And, as far as I know, it was not called “iWL pathway”.

Figs. 2, 3: what kind of growth is considered here? Autotrophic, heterotrophic? On which compounds? Fig. 3A2: what is 4HB bypath and where is it shown in the figure? Fig. 3A3: again, what is 4HB bypath, where 3HP/4HB cycle is shown, and why the reaction sequences called “partial rTCA” in Fig. 3A1 is called here “impaired rTCA”, and how can oTCA be integrated here?

p. 8: „In the type-A1 pathway, the complete WL pathway produces an ACOA influx, resulting in an OAA influx into the partial rTCA pathway. This is found for autotrophic methanogens such as *Methanococcus maripaludis* 22-24, which utilizes acetate that is converted into ACOA 22”: growth on acetate cannot be called “autotrophic”. What is meant here? That the addition of acetate makes the WL pathway redundant?

p. 8: “In the type-A2 and type-A3 pathways, the kinetic hypothesis would allow a complete rTCA cycle because the iWL pathway does not produce ACOA influx”: in the type A2 pathway, oxidative TCA is shown, at least in the Fig. 3A2.

p. 8: “The absence of PC caused an impairment in the reversal of the CS reaction, which was due to a decrease in the concentration of CIT and attributed to the significantly impaired right half of the

rTCA flux”: how the absence of PC can decrease citrate concentration?

p. 8: “Under chemolithoautotrophically growing conditions, the lack of PC yielded a partially impaired rTCA flux, whereas the corresponding Crenarchaeota, *Thermoproteus tenax*, and *Acidianus hospitalis* possess the dicarboxylate/4-hydroxybutyrate (4HB) pathway²⁶ and 3-hydroxypropionate/4-hydroxybutyrate (3HP/4HB) pathway²⁷, respectively; thus, these alternative carbon fixation pathways work to compensate for the partially impaired rTCA flux”: “Under chemolithoautotrophic growth conditions”. Thermoproteales do not possess PC but use PEP carboxylase instead, so the lack of PC is not of any importance here. Although, I am not sure that it’s appropriate to say “to compensate for the partially impaired rTCA flux” discussing a representative of microbial domain (Archaea), if no single archaeon was shown to use the rTCA cycle. There is nothing to compensate, they have other means for autotrophic growth.

p. 10: what is “nearly LUCA”?

p. 11: what is “4HB carbon fixation pathway”?

p. 11: “Subsequently, the heterotrophic anaerobic archaea would acquire enzymes necessary for a 4HB or 3HP/4HB carbon fixation pathway and secure the ability to grow not only heterotrophically, but also chemolithoautotrophically, as seen in A3”: Fig. 4A3 shows succinyl-CoA carboxylation to 2-oxoglutarate. Is any archaeon with the 3HP/4HB or DC/4HB cycle known to use this reaction?

Fig. 4A2: what is “incomplete 4HB bypath”?

Reply to the comments by Reviewer #1

We thank your constructive comments and useful suggestions. Reviewer's comments and our responses are listed below.

Reviewer's comment: 1) The authors also need to explain their method of estimating the free energy change values. What algorithm was used? What substrate concentrations or other physiological conditions were used?

Reply: We added the explanations and revised the descriptions in the text.

At the end of **Kinetic Network Model** in **Materials and Methods**, we added the following sentence.

The calculation of the apparent equilibrium constant and the apparent reaction Gibbs free energy is explained in **General aspects on kinetic modeling of enzymatic reactions** of the SI.

We revised **Simulations** in **Materials and Methods** as follows:

The steady-state concentrations and fluxes for all the organisms of which results were discussed in the main text and the SI were determined by the kinetic network model simulation based on Eqs. 1–16 in the SI. The concentrations of chemical species used as the model parameters in the kinetic network simulations were listed by Table A1 in the SI. Several model parameters in Table A1 were estimated so that the kinetic network model simulation reproduced the concentration of metabolites experimentally determined for *D. acetivorans* grown autotrophically (Table S2 in the SI).

Reviewer's comment: 2) Source 1 hypothesizes the genome of LUCA, as well as presents a hypothesis of its metabolism reconstructed from genomic data. If the goal is to highlight how the evolution of carbon metabolism occurs from LUCA, some schematic showing the different evolutionary pathways from LUCA's metabolism to that of modern species would be extremely helpful and instructive for readers.

Reply: We thank the comment and also agree with it. Our kinetic approach provides possible carbon metabolism flux patterns, while it cannot uniquely determine that of LUCA. Thus it is hard only for this approach to construct the tree of evolution pathway from LUCA to modern species. On the other hand, we can generate possible elementary evolution processes by linking two different flux patterns. Figure 4 summarizes the carbon metabolism flux patterns for deep-branching bacteria and archaea derived from the kinetic network model simulations with enzymes identified by the KEGG genome database. By utilizing Fig 4 with the kinetic hypothesis, the elementary evolution processes between the carbon metabolism flux patterns are discussed in the main text.

Reviewer's comment: 1) In the results section, the following statement seems to be poorly explained: "The reductive flux was increased by the ACOA influx because of the resulting OAA influx, whereas the oxidative flux was obviously suppressed by the conflict with the reductive flux." Why did the authors conclude that the increase in the reductive flux and suppression of oxidative flux is a result of the OAA influx? Is this a potential mechanism that the authors are proposing or is there any evidence that suggests this mechanism?

Reply: Thank you for the helpful comment. As you pointed out, the explanation was insufficient. The ACOA influx yields both large reductive and small oxidative flux. The former strengthens the reductive flux on the left side of rTCA cycle, while the latter is weakened by the competition with the reductive flux component. We revised the explanation on Fig. 1c as follows:

The reductive flux was increased by the ACOA influx because of an increase in the flux through the ACOA-PYR-OAA pathway, whereas the absolute value of oxidative flux from ACOA to AKG likewise caused by the ACOA influx was 5×10^{-7} times smaller than that of the reductive flux, upon comparison with the reductive flux (Fig. 1c). This is because the oxidative flux caused by the ACOA influx was weakened by the reductive flux.

Reviewer's comment: 2) A direct statement of "the kinetic hypothesis" would go a long way toward attempting to clarify what this work seeks to achieve, e.g. "the kinetic hypothesis we investigate in this study is ...".

Reply: Thank you for the useful comment. We added the following sentence above Fig. 1. In short, the kinetic hypothesis we propose in this study is that the complete rTCA cycle is never observed in single organisms because of being kinetically unstable by the competition between the reductive and oxidative flux if a carbon fixation pathway including the WL pathway yields ACOA influx into the rTCA cycle.

Reply to the comments by Reviewer #2

Dear Dr. Rogier Braakman,

We appreciate your careful reading of the manuscript, giving us constructive comments and useful suggestions, and encouraging its publication. According to the comments and suggestions, we revised the main text and the SI. Reviewer's comments and our responses are listed below.

Reviewer's comment: 1. I did not fully understand how the modeling approach works. A typical example of the general issues highlighted above comes from the following sentences in the first paragraph of the Results section "In the present study, a kinetic network model with enzymes identified in *T. takaii* was developed, and the feasibility of the chemolithoautotrophic rTCA cycle due to a reversal of the CS reaction was demonstrated (Fig. 1a). The obtained fluxes were consistent with the direction experimentally assigned by Nunoura et al., except for the fluxes between malate (MAL) and pyruvate (PYR) and between oxaloacetate (OAA) and PYR." So, over the course of just two sentences the text goes almost directly from mentioning what was done to the results, but for me it left many questions about the method unanswered:

Reply: Thank you for the useful comments on the explanation of the kinetic network model and the simulation method. As mentioned in the next reply, we revised the insufficient explanations in **Materials and Methods**.

Reviewer's comment: - What exactly do you mean by kinetic model in this case? It would really help the reader if you could explain at a high level how this approach works. Indeed, given that this modeling effort forms the heart of the whole paper, spending a full paragraph just explaining how it works would not be inappropriate. Even after reading the Methods section and Supplement it was still not entirely clear to me. I finally understood you built a model of 22 total reactions (when reading the main text, I first thought you built full genome-scale models), including a "biomass synthesis" reaction. Does the latter imply you took an FBA-type approach to the modeling?

Reply: Thank you for the useful comments. We checked the simulations section in **Materials and Methods** and found the following insufficient explanation.

The steady-state concentrations and fluxes for the kinetic network models of various organisms (Figs. S2 and S3 in the SI) including *T. takaii* and *D. acetivorans* were determined.

We revised this as follows:

The steady-state concentrations and fluxes for all the organisms of which results were discussed in

the main text and SI were determined by the kinetic network model simulation based on Eqs. 1–16. The concentrations of chemical species used as the model parameters in the kinetic network model were listed by Table A1 in the SI. Several model parameters in Table A1 were estimated so that the kinetic network model simulation reproduced the concentration of metabolites experimentally determined for *D. acetivorans* grown autotrophically (Table S2 in the SI)

Reviewer’s comment: - I also think I understood that the model was built on a foundation in which the free energies (and thus directionality) of reactions were treated as variables that depend on the concentrations of reactant, but it wasn’t clear to me how you then explored the model dynamics. That is, a single list of free energies of reaction was given in the supplement, but of course reaction free energies change depending on the concentrations of the reactants, which is part of what you are modeling, so how did the modeling work in practice? Did you take some kind of recursive modeling approach until the network equilibrates? And given that several simulations were done, with each solution presumably having their own Gibbs free energies of reaction, why were only single values listed?

Reply: As you mentioned, the reaction Gibbs free energy by Eq. A8 determines the direction of each enzymatic reaction, while we did not directly use it to simulate the carbon metabolism. Instead of Eq. A8, the apparent equilibrium constant Eq. A7 is used in the kinetic model of enzymatic reaction, which also includes total concentrations of reactants and products, indicating that the direction of enzymatic reactions depends on the concentrations of reactants and products. We added the following explanation at the end of first subsection “**General aspects on kinetic modeling of enzymatic reactions**” in the revised SI (page 4).

The apparent equilibrium constant Eq. A7 is used in the kinetic model of enzymatic reactions given below, so that the rate of enzymatic reactions is determined by the overall concentrations of reactants and products obtained from the binding polynomial Eq. A2. The direction of reaction determined by the kinetic models of (enzymatic) reactions (Eqs. B1–B17) depends on the concentrations of reactants and products and is consistent with the reaction Gibbs free energy given by Eq. A8. All the results discussed by the text and SI are based on the steady state concentrations and fluxes that are determined by the kinetic network model simulations based on the ordinary differential equations (1)–(16).

Reviewer’s comment:- Related to the previous question, I was generally a bit confused about how Gibbs free energies were used, since the text talks about “standard Gibbs free energies” (ΔG^0) which are for standard conditions (importantly including reactant concentrations of 1M, far above the more typical mM levels of cells), and the legend of Table S2 similarly seems to suggest the listed values are

standard Gibbs free energies of reaction $\Delta_r G^0$. However, equations A7-A9 suggest you are calculating the Gibbs free energies of reaction using physiologically relevant concentrations. But then this comes back to my previous point that each network solution presumably has its own set of Gibbs free energies, since concentrations are changing, making me wonder why only a single list of values is given in Table S2. It would be good if you make sure this is really clear throughout.

Reply: Equation A7 includes no concentrations of the reactants and products in enzymatic reaction except for H_2O and H^+ , thus the apparent reaction Gibbs free energy provided by Eq. A9 also includes no concentrations of the reactants and products. Therefore, the values provided by Table S3 in the revised SI (S2 in the previous version) do not depend on the concentrations of the reactants and products. On the other hand, the reaction Gibbs free energy provided by Eq. A8 depends on all the concentrations of the reactants and the products, thus, for instance, the reversal of citrate synthase reaction which requires the apparent reaction Gibbs energy $\Delta_r G_i^{tot}$ of +53.43 kJ/mol (Table S3 in the revised SI) is possible by high reactant and low product concentrations on the reversed reaction. If the steady state rTCA flux is determined to be zero by the kinetic network model simulation, the obtained steady state concentrations compensate the sum of $\Delta_r G_i^{tot}$ over the rTCA cycle, -108.76 kJ/mol, provided by Table S3 in the revised SI, so that the sum of reaction Gibbs free energy by Eq. A8 over the rTCA cycle becomes zero. We added the following explanation at the end of first subsection “**General aspects on kinetic modeling of enzymatic reactions**” in the revised SI (page 3–4).

It is noted that the apparent reaction Gibbs free energy $\Delta_r G^{tot}(T, I)$ by Eq. A9 is introduced to characterize the exergonicity of each enzymatic reaction. As seen in Table S3, even if the enzymatic reaction is significantly exergonic with huge negative value of $\Delta_r G_i^{tot}$, it can be reversal by high reactant and low product concentrations of the reversed reaction, as shown by Eq. A8 of the reaction Gibbs free energy.

Reviewer’s comment:- Stepping back, did you solve for the basic ability to grow (i.e. can biomass be produced yes/no) and then examine fluxes that emerge in the model solution, or did you systematically vary metabolite concentrations to see which combinations allowed growth? Relatedly it was also not clear to me which metabolite concentrations you fixed and which you allowed to vary, and for those that you allowed to vary how their ranges were constrained to maintain physiological realism.

Reply: The concentrations of chemical species listed in Table A1 are set to be constant, while the concentrations of the other metabolites involved in the kinetic network model are determined as the steady-state concentrations depending on the fixed concentration values of the chemical species. Simultaneously, we obtain the steady-state fluxes of biomass synthesis (Eq. B18 in the revised SI), which are related to a cell growth ability. We added these explanations in **Materials and Methods**. In

addition, we investigated oxidized ferredoxin (Fdx_{ox}) concentration dependence of chemolithoautotrophic biomass synthesis (Fig. S2 in the revised SI) and CO₂ concentration dependence of biomass synthesis rate (Fig. S3 in the revised SI). We added the following discussions at page 4 in the main text.

It was also confirmed that the autotrophic growth simulated here was robust for extensively varying the ratio of the concentrations of reduced ferredoxin (Fdx_{red}) and oxidized ferredoxin (Fdx_{ox}) (Fig. S2 in the SI). Recently, it was experimentally observed that high partial pressure of CO₂ drove the rTCA cycle with the reversal of the CS reaction. Our kinetic network model reproduced the CO₂-dependence of autotrophic growth rate observed for *D. acetivorans* (Fig. S3 in the SI).

Reviewer's comment: 2. The kinetic hypothesis states that because influx of acetyl-CoA acts to inhibit reductive flux through the TCA cycle, the linkage between rTCA and WL is disfavored. But, organisms with the WL pathway generally redirect acetyl-CoA/acetate flux toward other endpoints that are released from the cell, e.g. acetate → methane (methanogens) or acetyl-CoA → acetate → excretion (acetogens), which indicates that acetyl-CoA might accumulate much less in those cases than is suggested in the text here. Did you explore this possibility? I couldn't tell if the kinetic model here has the possibility of such excretion mechanisms? If you added them, would it affect your results and therefore your conclusions/hypothesis?

Reply: Thank you for the thoughtful comment. The kinetic network model basically includes outflux of ACOA as the biomass synthesis (Eq. B18 in the revised SI). On the other hand, we did not introduce an additional outflux of ACOA as an excretion pathway in the kinetic model simulations for methanogens (Figs. S5 b–e in the revised SI) and acetogens (Figs. S4 a, b in the revised SI). However, as shown in the SI figures 4 and 5, these organisms cannot have a complete rTCA cycle because of the lack of several enzymes for the rTCA cycle. This observation does not contradict with the kinetic hypothesis we proposed: a complete rTCA cycle is never observed in organisms because of its kinetic instability from the competition between the reductive and oxidative flux, as long as a carbon fixation pathway including the WL pathway yields ACOA influx into the rTCA cycle; unless the ACOA influx disappears, the gene for the enzyme working at the less active reaction on the rTCA cycle is lost or that of enzymes lacked on the less active part is not newly gained during evolution of LUCA.

Reviewer's comment: - Relatedly, the text states that simulated concentrations of key metabolites are consistent with concentrations measured experimentally. However, looking at Table S1, while concentrations of citrate, succinate and malate are indeed relatively similar between model and experiment, the concentration of acetyl-CoA is nearly an order of magnitude higher in the model than in the experiment, and it is acetyl-CoA that is the key intermediate in your argument for a kinetic

conflict between the rTCA cycle and WL pathway. This discrepancy and how it affects simulations under enhanced acetate/succinate influx might deserve further comment. Are you sure it does not affect the results?

Reply: Thank you for the careful reading. As you pointed out, the concentration of ACOA obtained from the model is higher than the experimental value. As for the acetate uptake, we discussed two cases with the large and small ACOA influx (Fig. 1c and 1d). These results, at least, indicate that an additional influx of ACOA, in other words, the coexistence of the WL and rTCA pathways adversely affects the maintenance of complete rTCA flux. We revised the related discussions at page 5 as follows.

The reductive flux was increased by the ACOA influx because of an increase in the flux through the ACOA-PYR-OAA pathway, whereas the absolute value of oxidative flux from ACOA to AKG likewise caused by the ACOA influx was 5×10^{-7} times smaller than that of the reductive flux, upon comparison with the reductive flux (Fig. 1c). This is because the oxidative flux by the ACOA influx was weakened by the reductive flux. Furthermore, a smaller ACOA influx did not result in a small oxidative flux but instead impaired the reductive flux from AKG toward both OAA and ACOA (Fig. 1d). This is because the smaller oxidative flux caused by the smaller ACOA influx was slightly overcome by the reductive flux. The absolute value of the impaired reductive flux was $\sim 1 \times 10^{-5}$ times smaller than that of the reductive flux on the left side of rTCA cycle due to the competition with the oxidative flux component.

Reviewer's comment: - did you perform kinetic modeling runs for these networks as in the previous section, or did you do metabolic network reconstructions from genomes, or both? The discussion of "obtained fluxes" throughout this section implies numerical modeling was done, while the use of KEGG implies network reconstructions. If the work done here was solely network reconstruction, it would be best if the text avoids use of "fluxes" and "flux patterns" to describe the results, since those suggest modeling results. This phrasing in terms of fluxes is also used in the previous section where modeling was done, and both sections show schematic flux diagrams as the main results, which is partly where the confusion comes from. If modeling and calculations were in fact done (or also done, alongside network reconstructions), then the results should be listed somewhere beyond only the schematic diagrams. The supplement currently only contains one network variant (that from the first section), so if other variants were also modeled those results should also be shown somewhere.

Reply: We performed the kinetic network model simulations for all bacteria and archaea shown in the revised SI Figs. S4 and S5, where enzymes identified by the KEGG genome database were incorporated into the kinetic network models. The purpose of performing the kinetic network model simulations for bacteria and archaea we considered here is to confirm whether or not reductive or

oxidative fluxes actually occur on the carbon metabolic network reconstructed by the KEGG genome database under the autotrophic growth condition (listed by Table A1 in the revised SI). We observed the partial reductive and oxidative flux on the TCA pathway for two acetogens (Figs. S4a and S4b in the revised SI). In addition, we found that the necessity of Pyruvate Carboxylase (PYR→OAA) to maintain rTCA flux (Fig. S4d in the revised SI), while that is not necessarily needed for producing oTCA flux under heterotrophic growth conditions (Fig. S5f in the revised SI). These findings based on the kinetic network model simulations, which were shown in the SI, were argued in the main text. To show several typical numerical results of the steady state fluxes determined by the kinetic network simulation, we added the steady state flux values in the revised SI Table S1a and S1b.

Reviewer's comment:- If the work indeed consisted of metabolic network reconstructions, more details on the approach should be given, at the very least in the Methods section. Was it done via web-browsing on the various organism pages in the KEGG database? Or did you perform BLAST searches of reference genes/enzymes in the given organisms? It should be noted that reconstructions based on web-browsing of KEGG has some uncertainties due to misannotation of genes. It is therefore generally advisable to follow up web-browsing to get first impression with more careful sequence-based searches and analyses. I cannot quite tell which approach was used in this case.

Reply: Thank you for the kind explanation on metabolic network reconstructions. Unfortunately, we have no skill on sequence-based searches and analyses yet. We added the following sentence in **Materials and Methods**.

The information on the gene of enzymes was basically obtained by web-browsing on specific organism pages in the KEGG database.

Reviewer's comment: 4. On a more conceptual side, I also didn't fully understand aspects of the discussion around evolution of pathway variants. The paper concludes there is a kinetic conflict between the rTCA cycle and WL pathways that prevents both from co-occurring in the same organism, which is an interesting conclusion if it holds up around clarifications mentioned above. However, much of discussion in the final two parts of the result section (i.e. the KEGG-based surveys of bacteria/archaea) and the discussion section focuses on the "avoidance" of this conflict over the course of evolution. I'm not sure I understood this point, as it implies that there is in fact some other selective pressure acting in parallel that favors the combination of the rTCA cycle and WL pathway. If there isn't some intrinsic benefit to the direct linkage of the full rTCA cycle and WL pathway, why and how would a conflict ever arise? Indeed, use of the term "conflict" implies there are two forces working opposition to each other, one in favor of linking the pathways, the other in favor of breaking that linkage. The only other scenario in which I can see "evolutionary avoidance" of the kinetic conflict

coming into play is if at some point in early evolution the two pathways did co-occur in the same cell, in which case evolution away from that state (at least partly driven by the need to remove the previous conflict) could have created all the disconnected variants seen today.

Reply: Thank you for the thoughtful comments. We understood the possibility that there would be some other selective pressure acting in parallel that favors the combination of the rTCA cycle and WL pathway. At the beginning of the discussion, we revised the related sentence as follows.

In the pioneering work by Braakman and Smith, a single connected redundant network consisting of a complete WL pathway and a full rTCA cycle was proposed as a more robust and plausible topology that (pre-) LUCA should possess on the basis of phylometabolic analysis. In contrast, the results obtained from our kinetic network model do not support their prediction on the coexistence of those pathway in one organism, unless other predominant selective pressure that favors the combination of those pathways exists in parallel.

Reviewer's comment: - Directly related to the previous point, the text states in the first paragraph of the Discussion section that the results obtained here are inconsistent with the conclusions in Braakman and Smith 2012, which I also didn't fully understand. I generally don't bring my own research directly into the review process, but since the text directly addresses our paper and I happen to be reviewing this paper, I will briefly respond. In discussions on the early evolution of metabolism it is important to distinguish between two major phases of evolution: 1) pre-LUCA evolution, which reaches back to the origin of life and proceeds via pre- and proto-cellular stages of chemical self-organization to the emergence of fully integrated cells, and 2) post-LUCA evolution, which involves evolutionary diversification away from the LUCA after cellular life was robustly established. Like the current paper, our 2012 paper reconstructs evolution of CO₂-fixing pathways back from extant life to the LUCA, in that case identifying an integrated rTCA+WL pathway as the most likely ancestral CO₂-fixing pathway. However, the subsequent statements about the "more robust and plausible network topology" of this network refer to pre-LUCA evolution. That is, we argued that the rTCA+WL network would have benefited from a built-in network-topological robustness during the process of chemical self-organization leading from prebiotic chemistry to fully integrated cellular life. Once fully integrated cells had emerged, that network-topological robustness would have become redundant, and, indeed, post-LUCA evolution produced a number of variants in which the rTCA and WL pathways have become disconnected from one another. The current paper similarly examines post-LUCA evolution, since it surveys diversity in extant metabolisms, so it's not totally clear to me why the results are inconsistent with what we found in our paper. Indeed, the papers actually seem highly compatible, with the main difference lying in the emphasis on different (but again compatible) forces driving the disconnection between the rTCA cycle and WL pathways. Anyway, this is not meant to push the views

of our paper – the authors should obviously feel free to discuss their results and speculate as they see fit, and indeed it is good for discussion, debate and disagreement to continue in the larger community - but i did want to highlight how it is important to distinguish between pre- and post-LUCA evolutionary processes when discussing most likely properties of the LUCA. LUCA was by definition the product of pre-LUCA evolution, and so if the current paper wants to draw conclusions about what was the most likely pathway in the LUCA, it would be good to explicitly consider how and why forces found to have shaped post-LUCA evolutionary diversity should be extended back to understanding evolutionary processes in pre-LUCA times.

Reply: We understood that it was important to distinguish between pre-LUCA and post-LUCA evolution in the discussions of early evolution of metabolism. Below the mention about the prediction by Braakman and Smith (at the beginning of the discussion), we added the following sentences.

The inconsistency between their and our predictions should be due to whether or not the kinetic factors are taken into consideration. On the other hand, the arguments on post-LUCA evolution by them seem not to be inconsistent with our kinetic hypothesis.

Reviewer's comment: 5. A somewhat smaller point, but i just wanted to note that in the Discussion section there is discussion of evolutionary processes in terms of various lineages "trying to acquire" or "securing" various enzymes. This phrasing endows microbes with a lot of agency and foresight, and gives the impression that cells are able to "choose" their evolutionary trajectories. I'm not sure this was intentional? It might be good to double check the language used to discuss evolutionary dynamics.

Reply: Thank you for the helpful comments. We revised "try to acquire" and most of "acquire" as "gain" and "secure" as "gain".

Reviewer's comment: - first paragraph of results, middle of p. 3: What do you mean by "feasibility was demonstrated"? The text only cites Fig. 1A for this claim, but it shows only a schematic diagram. Where are the actual results shown? What were your key assumptions and how did you test if the citrate synthase reaction could be reversed? Did you vary concentrations, or something else? This should all be explained more clearly, see previous comments

Reply: Thank you for the helpful comments. We added the steady-state fluxes obtained from the kinetic network model for *T. Takaii* grown chemolithoautotrophically in the revised SI Table S1a. We also confirmed that the simulated autotrophic growth was robust for varying the ratio of the concentrations of reduced ferredoxin (Fdx_{red}) and oxidized ferredoxin (Fdx_{ox}) (Fig. S2 in the revised SI).

Reviewer's comment:- also first paragraph of results, middle of p. 3: What do you mean by the “fluxes were consistent” with previous experimental results? Do you mean just direction, or also magnitude? Please clarify in the text.

Reply: The experiments determined the direction of carbon fluxes. The directions of obtained fluxes were consistent with the experimental observations. We revised all the related sentences in the text (page 3–4).

Reviewer's comment:- also first paragraph of results, middle of p. 3: The text contains the statement that the direction of two fluxes in the TCA cycle (between MAL and PYR and between OAA and PYR) found in the simulations are not consistent with the experimental findings of Nunuora et al. Does this not invalidate the conclusion? I don't understand how full rTCA cycling can be maintained if the flux of two reactions is reversed. It would be great if this could be addressed a bit more.

Reply: Thank you for careful reading and crucial comments. “between OAA and PYR” is mistake. We revised “between OAA and PYR” as “between OAA and phosphoenolpyruvate (PEP)”. In addition, we checked that the gene knockouts of malic enzyme between MAL and PYR and/or of Phosphoenolpyruvate Carboxykinase between OAA and PEP held a complete rTCA flux, indicating that these fluxes are not essential for maintaining the rTCA cycle (shown in the revised SI Table S1b). Here, the gene knockout means that the simulations were performed without corresponding enzymes.

Reviewer's comment:- bottom of p. 3/top of p.4: the text states that the rTCA flux was evaluated under various gene knockouts. How was this done? I cannot find any details. The text also states that results are listed in Table S3, but table S3 only contains text entries that flux was impaired/reduced/lost. Actual results again seem to be missing.

Reply: The gene knockout means that the simulations were performed without corresponding enzymes. We added and revised the explanation on the gene knockout simulations for the revised SI Table S4 (S3 in the previous version).

Reviewer's comment:- section on TCA cycle bifurcation, middle p. 4: can you elaborate a bit on the underlying molecular mechanisms that drive the bifurcations? Do concentrations of some intermediates increase more than others, thereby causing the flux to reverse? In principle one could imagine influx of acetate/succinate simply driving the rTCA cycle forward, so it would be great to have a bit further discussion, since modeling studies like in this paper seem ideally suited to

disentangle the subtleties of the dynamics.

Reply: As you pointed out, the influxes of acetate and succinate (SUC) increase ACOA and SUC concentrations, respectively, thus turn over the sign of the reaction Gibbs energy (given by Eq. A8 in the SI) for enzymatic reactions in the TCA cycle, so that the fluxes simulated by the kinetic network model (Eqs. 1–16 in the SI) are reversed.

Reviewer's comment:- section on TCA cycle bifurcation, middle p. 4: Like in the previous section, the text here states that the direction of two fluxes in the TCA cycle (between MAL and PYR and between OAA and PYR) found in the simulations are not consistent with the experimental findings in T. takaii. Does this again not invalidate the conclusion? See previous comment...

Reply: As mentioned above, we revised “between OAA and PYR” as “between OAA and phosphoenolpyruvate (PEP)”. In addition, we checked that the gene knockouts of malic enzyme between MAL and PYR and/or of Phosphoenolpyruvate Carboxykinase between OAA and PEP held a complete rTCA flux (shown by the revised SI Table S1b), indicating that these fluxes are not essential for maintaining the rTCA cycle.

Reviewer's comment:- section on surveys in other bacteria/archaea, top of p. 6: why were cyanobacteria first excluded and then reincorporated into the analysis? This was not clear from the text

Reply: Figure 2 shows the results for deep-branching bacteria simulated by the autotrophic growth condition (Table A1 in the revised SI). Cyanobacteria was here excluded, because it has an oTCA cycle and that was simulated with a lower ratio of reduced agent to oxidized agent, while was later incorporated into the argument.

Reply to the comments by Reviewer #3

We appreciate your careful reading of the manuscript, giving us constructive comments and useful suggestions, and encouraging its publication. Reviewer's comments and our responses are listed below.

Reviewer's comment: Starting with the claim that the rTCA cycle and the WL pathway belong to the most ancient metabolic pathways, the authors analyze a possible co-occurrence of these two pathways in LUCA, and this causes a question how the authors actually came to the idea to analyze it. There is no single organism that has both pathways at once, and the reason for having them in one cell is at best not obvious. In fact, the co-existence of two pathways in one organism was not unequivocally demonstrated in any organism in general, although there are good reasons to believe that this is the case for the Calvin cycle and the rTCA cycle in some sulfur bacteria. The proposal of Braakman and Smith of the co-functioning of WL and rTCA in LUCA that justifies analysis is mentioned first in the discussion. I think that it has to be mentioned and briefly discussed in the Introduction, in order to avoid the misunderstanding.

Reply: Thank you for the useful and constructive comments. We added the following sentences into the final paragraph of the introduction.

In the pioneering work by Braakman and Smith, a single connected redundant network consisting of a complete WL pathway and a full rTCA cycle was proposed as a more robust and plausible topology that (pre-)LUCA should possess. However, our kinetic results do not support their prediction on the coexistence of those pathway in one organism, unless other predominant selective pressure that favors the combination of those pathways exists in parallel.

Reviewer's comment: p. 2: „Currently, six different autotrophic pathways responsible for carbon fixation are recognized“: with the glycine reductase pathway, seven (DOI: 10.1038/s41467-020-18906-7).

Reply: Thank you for the informative comment. We revised the text as follows.

Currently, there are seven known different autotrophic pathways responsible for carbon fixation, including the newly demonstrated reductive glycine (rGly) pathway

Reviewer's comment: p. 2: “In the present study, kinetic simulations of the autotrophic rTCA cycle for these bacteria are presented, and the mechanism by which the CS reaction is reversed, which was thought to function only in the oxidative direction, is revealed in dynamical systems”: you may wish to discuss the recent paper of Steffens et al showing that high CO₂ partial pressure allows CS to be

reversed (doi: 10.1038/s41586-021-03456-9).

Reply: Thank you for the useful information. We calculated CO₂ concentration dependence of biomass synthesis rate (Fig. S3 in the revised SI) and added the following comments at the first subsection of the result (page 4) in the main text.

Recently, it was experimentally observed that high partial pressure of CO₂ drove the rTCA cycle with the reversal of the CS reaction. Our kinetic network model reproduced the CO₂-dependence of autotrophic growth rate observed for *D. acetivorans* (Fig. S3 in the SI).

Reviewer's comment: p. 2: „direct connection of the WL pathway to the rTCA cycle”: I am not sure what is meant here. What is this “direct connection”, taking into account that the WL pathway and the rTCA cycle both have the same product (acetyl-CoA)?

Reply: Thank you for the crucial comment. We revised this as “the coexistence of a WL pathway with a rTCA cycle”, and also revised all the similar expressions in the text.

Reviewer's comment: p. 3: “In the present study, a kinetic network model with enzymes identified in *T. takaii* was developed, and the feasibility of the chemolithoautotrophic rTCA cycle due to a reversal of the CS reaction was demonstrated (Fig. 1a)”: I am not sure how this model works. What comes in and what comes out? What is included in the model? What are the premises and assumptions? Is autotrophic growth meant here? Please explain it in details.

Reply: Thank you for the helpful comment. We added the following explanation at page 3.

To investigate the direction of carbon metabolic fluxes on the network, the steady-state fluxes were determined by the kinetic network model (Eqs. 1–16 in the SI) with the fixed concentrations of chemical species listed in the SI Table A1. The five universal precursors of anabolism, ACOA, pyruvate (PYR), phosphoenolpyruvate (PEP), oxaloacetate (OAA), and 2-oxoglutarate (AKG: α -ketoglutaric acid), are assumed to be consumed by biomass synthesis in the kinetic network model.

Reviewer's comment: p. 3: “In addition, the ferredoxin-dependent enzymes working in the rTCA cycle involve two reduced/oxidative ferredoxins in the reductive reactions (Table S2 in the SI); thus, the concentration ratio between reduced and oxidative ferredoxins needed to overcome the energetically unfavorable reaction is lower than that with NADH/NADPH”: what is “oxidative ferredoxin”? Table A1 shows certain values for concentrations of reduced and oxidized ferredoxin. Where do these values come from? It's claimed that they are “estimated for rTCA cycle of *T. takaii*”. Where and how were they estimated? The given ratio Fd_{red}/Fd_{ox} is 50,000, basically meaning that

ferredoxin is reduced to 100,00% in the cell. How could it be achieved?

Reply: Thank you for the helpful comments. We revised “oxidative ferredoxin” as “oxidized ferredoxin”.

The ratio $Fd_{red}/Fd_{ox} = 50,000$ was estimated so that the kinetic network model reproduced the concentrations experimentally determined for chemolithoautotrophically grown *D. acetivorans* (Table S2 in the revised SI). We additionally confirmed that the simulated autotrophic growth was robust for varying the ratio of the concentrations of reduced ferredoxin (Fd_{red}) and oxidized ferredoxin (Fd_{ox}) (Fig. S2 in the revised SI).

Reviewer’s comment: p. 3: “It is noted that *D. acetivorans* possesses a serine synthesis pathway that results in a PYR influx into the rTCA cycle, whereas *T. takaii* does not (Figs. S2e and S2fin the SI)”: what is “serine pathway”? The mechanisms that (in my imagination) can allow “pyruvate influx” in an autotrophic bacterium in the absence of pyruvate synthase are anaplerotic reactions functioning during growth on acetate (the glyoxylate cycle, the ethylmalonyl-CoA pathway). However, I believe that it is not what is meant here?.

Reply: Thank you for the useful comment. In the KEGG database of carbon metabolism for *Desulfurella acetivorans* (<https://www.kegg.jp/pathway/dav01200>), we found an enzyme converting serine to pyruvate, while we did not for *Thermosulfidibacter takaii* (<https://www.kegg.jp/pathway/ttk01200>). It is called serin metabolism, whereas the mechanism converting serine to pyruvate is unclear. We used *T. takaii*-type and *D. acetivorans*-type models to investigate the role of pyruvate synthase and pyruvate carboxylase on the rTCA cycle. Here, we assumed that the *D. acetivorans*-type model possessed a PYR influx into the rTCA cycle via serine (SER) metabolism, whereas *T. takaii*-type model did not. The difference between these models revealed the importance of pyruvate carboxylase for maintaining the rTCA cycle, as argued in Table S4 in the revised SI.

Reviewer’s comment: p. 4: “In contrast, the model of *D. acetivorans* with the gene knockout of pyruvate synthase maintained the rTCA flux, whereas there was a decrease with *T. takaii*”: how is it possible? What is the difference between *T. takaii* and *D. acetivorans* in this regard? In fact, the mentioned Steffens et al paper highlights the essentiality of the pyruvate synthase reaction in *D. acetivorans*.

Reply: In the model of *D. acetivorans* with the gene knockout of pyruvate synthase, PYR is not yielded via the conversion of ACOA by pyruvate synthase, while the PYR influx is assumed from

serine metabolism, thus PYR is converted to OAA by pyruvate carboxylase. As a result, the reduction in the rTCA flux by the KO of pyruvate synthase was not so larger than the model of *T. takaii*. However, it is not obvious whether or not the differences in the rTCA fluxes between *T. takaii* and *D. acetivorans* are experimentally observed by the gene knockout of pyruvate synthase because the effects of PYR influx via serine metabolism on *D. acetivorans* are uncertainty. Even though, the investigation of the role of PC and PS by utilizing the *T. takaii*-type and *D. acetivorans*-type models was useful since it shed light on the importance of PC in the rTCA cycle. This discussion is presented at the end of the first subsection in the result (page 4–5).

Reviewer’s comment: In the manuscript, the importance of pyruvate carboxylase is discussed. However, another widespread enzyme catalyzing C3 conversion to C4, PEP carboxylase, is not even mentioned. PEP carboxylase is probably not the best choice for bacteria using citrate synthase version of the rTCA cycle, but this enzyme does function in many organisms for autotrophic CO₂ fixation. In addition, malic enzyme can actually also work backwards, being an attractive alternative to the PC reaction.

Reply: In the present model, we confirmed that no significant effects on the rTCA flux were observed by the gene knockout of PEP carboxylase or/and of malic enzyme, as shown by Table S1b in the revised SI. This was argued in the first subsection of the result (page 3–4).

Reviewer’s comment: p. 4: “The obtained results were consistent with the experimental observations for *T. takaii* grown chemolithomixotrophically with SUC, except for the fluxes between MAL and OAA and between OAA and PYR”: could you please explain what does consistent exactly mean in this case, and what is wrong with fluxes between malate, oxaloacetate and pyruvate. The same for acetate.

In Fig. 1b, CS is shown to function backwards during growth on succinate in the figure. What are evidences for it?

Reply: To explain clearly, we revised the explanation as follows.

The directions of obtained fluxes were consistent with the experimental observations for *T. takaii* grown chemolithomixotrophically with SUC, except for the directions between MAL and OAA and between OAA and PYR.

The directions of obtained fluxes were consistent with the experimental observations for *T. takaii* grown chemolithomixotrophically with acetate, which is converted into ACOA, except for the directions between SCOA and AKG and between OAA and PEP.

The autotrophic growth condition same as Fig. 1a (listed by the Table A1 in the SI) was used in Figs. 1b-1d, thus the reversal of CS reaction was possible in the presence of SUC. The directions of the fluxes shown in Fig. 1b are consistent with those experimentally determined for *T. takaii* grown chemolithomixotrophically in the presence of succinate (Nunuora et al., 2018).

Reviewer's comment: Figs. 1c, 1d: the flux only in the direction of acetyl-CoA carboxylation is shown. However, if I understand the Nunuora et al (2018) data correctly, the presence/dominance of 4x13C in Glu after growth on fully labelled acetate suggest the functioning of CS in oxidative direction under these conditions. Also, the CS activity was very high on acetate, and the functioning of oxidative TCA cycle appears to be unavoidable. Doesn't the model contradict to the experimental data?

Reply: The oxidative flux in the CS reaction caused by an ACOA influx (Fig. 1c) under the autotrophic growth condition (listed in Table A1) is consistent with that for *T. takaii* grown chemolithomixotrophically in the presence of acetate (Nunuora et al., 2018). On the other hand, as shown in Fig. 1d, the reductive flux in the CS reaction was obtained from a smaller ACOA influx under the same autotrophic growth condition, because the reductive flux overcame a smaller oxidative flux caused by the smaller ACOA influx. This observation is consistent with that for *D. acetivorans* grown chemolithomixotrophically in the presence of acetate (Mall et al., 2018).

Reviewer's comment: p. 4: "the ACOA influx that was smaller than that used in Fig. 1c": could you please define this "smaller"?

Reply: Thank you for the helpful suggestion. We revised it as "smaller ACOA influx".

Reviewer's comment: p.4: "This result was consistent with the experimental observation for *D. acetivorans* grown chemolithomixotrophically in the presence of acetate, where the majority of acetate uptake was mainly transformed into PYR and incorporated into the rTCA cycle via the abovementioned pathway from PYR to OAA (see Fig. S1ain the SI)": if I understand it correctly, Fig. S1a shows autotrophic growth of *D. acetivorans*. Acetate growth is shown in Fig. S1b, and it proceeds via the oxidative, and not reductive TCA cycle. In fact, CS would not work in the direction of citrate cleavage in the presence of acetate/acetyl-CoA. This is simply the results of CS equilibrium, and additional "kinetic hypothesis" appears to be redundant here.

Reply: Figure S1b shows that the kinetic network model with low concentrations of reduced ferredoxin (Fdx_{red}) reproduces oTCA fluxes for *D. acetivoransi* grown heterotrophically in the presence of acetate (Mall et al., 2018). On the other hand, figure S1a shows the kinetic necessary

condition for maintaining rTCA flux under autotrophic growth conditions with high concentrations of reduced ferredoxin, and schematically illustrates the necessity of pyruvate carboxylase. As you pointed out, the kinetic hypothesis is interpreted as the effects of acetate/acetyl-CoA influx on the reversal of CS reaction under autotrophic growth conditions, thus is also related to CS equilibrium in the view of physical chemistry. On the basis of the kinetic hypothesis, we can argue the biological reason on the absence of coexistence of the WL and rTCA pathways in one organism, and also on the presence of TCA variants in the presence of the WL pathway.

Reviewer's comment: p. 4: “direct connection is disconnected”: ?

Reply: We revised “unless the direct connection is disconnected” as “unless the ACOA influx disappears”.

Reviewer's comment: p. 6: “The results for bacteria belonging to Firmicutes, Cyanobacteria, Aquificae, and Proteobacteria, which are expected to be close to LUCA (Ref. 15,17)”: in fact, the references 15 and 17 does not discuss LUCA at all. I am also wondering how the proposed close connection of e.g. Cyanobacteria to LUCA is compatible with LUCA having the WL pathway and the rTCA cycle. The same is true for p. 7, concerning Archaea.

Reply: Thank you for the comments on the connections to LUCA. As you pointed out, the connections of these bacteria and archaea to LUCA are unclear. Thus, we removed the mentions on the relation of these bacteria and archaea to LUCA.

Reviewer's comment: p. 6: “whether the WL pathway connects the TCA cycle, resulting in a direct ACOA influx”: I am not sure what is meant here.

Reply: We revised “whether the WL pathway connects the TCA cycle” as “whether the WL pathway coexists with the TCA cycle”.

Reviewer's comment: p. 6: incomplete WL pathway: what do you mean? Some of the enzymes of the pathway are involved in C1 metabolism in many (most?) organisms; however, this is not called as the WL pathway. The synthesis of serine (that is shown in Fig. 2 for this “iWL pathway”) requires glycine. Where does it come from? Or, do you mean the glycine synthesis via the glycine cleavage system? The corresponding pathway leading to serine was indeed discussed on several occasion and reconstructed in several metabolic engineering studies but has never been not shown (until now?) for natural CO₂ fixation. And, as far as I know, it was not called “iWL pathway”.

Reply: Thank you for the useful comment. As you pointed out, such an iWL pathway does not necessarily yield an SER influx, thus we revised this. Actually, “incomplete WL pathway” seems not to be generally used, while it is sometime referred to as. In the text, we mentioned as follows: “iWL pathway” also includes the cases where enzymes acting in the WL pathway do not mostly exist, simply indicating that there is no ACOA influx from outside. In addition, we removed all the arrows toward SER from the iWL pathway in the figures of both the text and SI.

Reviewer’s comment: Figs. 2, 3: what kind of growth is considered here? Autotrophic, heterotrophic? On which compounds? Fig. 3A2: what is 4HB bypath and where is it shown in the figure? Fig. 3A3: again, what is 4HB bypath, where 3HP/4HB cycle is shown, and why the reaction sequences called “partial rTCA” in Fig. 3A1 is called here “impaired rTCA”, and how can oTCA be integrated here?

Reply: Thank you for the useful comments. In Fig. 2, the autotrophic growth condition with the high Fdx_{red}/Fdx_{ox} (Table A1 in the revised SI) is assumed for the bacteria B1–B4 grown autotrophically. In Fig. 3, the autotrophic growth condition with the high Fdx_{red}/Fdx_{ox} (Table A1 in the revised SI) is assumed for A1 and A3 grown autotrophically, while a heterotrophic growth condition with a lower Fdx_{red}/Fdx_{ox} is assumed for A2 grown heterotrophically. In Figs. 3A2 and 3A3, we revised “4HB bypath” as “dicarboxylate/4-hydroxybutyrate (DC/4HB) pathway” and “dicarboxylate/4-hydroxybutyrate (DC/4HB) pathway” as “3-hydroxypropionate/4-hydroxybutyrate (3HP/4HB) pathway”. In Fig. 3A3, the DC/4HB and 3HP/4HB pathways are modeled as a combination of additional ACOA influx and the conversion of succinyl-CoA (SCoA) to ACOA. We added the following explanation for Fig. 3A3 at page 9–10 in the main text:

These alternative carbon fixation pathways yield an ACOA influx under chemolithoautotrophic growth conditions, thus, impairs the rTCA flux on the right-hand side, in the same manner as the WL pathway (Fig. 3A3).

Reviewer’s comment: p. 8: „In the type-A1 pathway, the complete WL pathway produces an ACOA influx, resulting in an OAA influx into the partial rTCA pathway. This is found for autotrophic methanogens such as *Methanococcus maripaludis* 22–24, which utilizes acetate that is converted into ACOA 22“: growth on acetate cannot be called “autotrophic”. What is meant here? That the addition of acetate makes the WL pathway redundant?

Reply: Thank you for pointing out the contradiction. We checked the literature and confirmed that “which utilizes acetate that is converted into ACOA” was incorrect, thus we removed this in the text.

Reviewer's comment: p. 8: "In the type-A2 and type-A3 pathways, the kinetic hypothesis would allow a complete rTCA cycle because the iWL pathway does not produce ACOA influx": in the type A2 pathway, oxidative TCA is shown, at least in the Fig. 3A2.

Reply: To avoid such a misleading, we revised this as follows.

In the type-A2 pathway, the absence of WL pathway does not prevent a complete rTCA cycle under autotrophic growth conditions. Nevertheless, the rTCA cycle is kinetically inhibited because of the absence of PC in this pathway.

Reviewer's comment: p. 8: "The absence of PC caused an impairment in the reversal of the CS reaction, which was due to a decrease in the concentration of CIT and attributed to the significantly impaired right half of the rTCA flux": how the absence of PC can decrease citrate concentration?

Reply: To explain this, we revised the explanation at page 9 as follows:

This absence led to the loss of the OAA influx from PYR into the rTCA cycle, thus causing a decrease in the concentration of CIT, which is the product of rTCA cycle, and resulted in the impairment of the reversed CS reaction.

Reviewer's comment: p. 8: "Under chemolithoautotrophically growing conditions, the lack of PC yielded a partially impaired rTCA flux, whereas the corresponding Crenarchaeota, *Thermoproteus tenax*, and *Acidianus hospitalis* possess the dicarboxylate/4-hydroxybutyrate (4HB) pathway²⁶ and 3-hydroxypropionate/4-hydroxybutyrate (3HP/4HB) pathway²⁷, respectively; thus, these alternative carbon fixation pathways work to compensate for the partially impaired rTCA flux": "Under chemolithoautotrophic growth conditions". Thermoproteales do not possess PC but use PEP carboxylase instead, so the lack of PC is not of any importance here. Although, I am not sure that it's appropriate to say "to compensate for the partially impaired rTCA flux" discussing a representative of microbial domain (Archaea), if no single archaeon was shown to use the rTCA cycle. There is nothing to compensate, they have other means for autotrophic growth.

Reply: Thank you for the useful comments. We revised the argument on Fig. 3A3 at page 9 as follows:

As shown in the type-A3 pathway, the partially impaired rTCA flux was observed even in the absence of complete WL pathway. In addition to the absence of PC, the type-A3 Crenarchaeota, e.g., *Thermoproteus tenax* and *Acidianus hospitalis*, possess the dicarboxylate/4-hydroxybutyrate (DC/4HB) pathway and 3-hydroxypropionate/4-hydroxybutyrate (3HP/4HB) pathway, respectively. These alternative carbon fixation pathways yield an ACOA influx under chemolithoautotrophic growth conditions, thus, impairs the rTCA flux on the right-hand side, in the same manner as the

WL pathway (Fig. 3A3).

Reviewer's comment: p. 10: what is “nearly LUCA”?

Reply: Since “nearly LUCA” is unclear, we removed all “nearly LUCA” in the text.

Reviewer's comment: p. 11: what is “4HB carbon fixation pathway”?

Reply: We revised “4HB carbon fixation pathway” as “DC/4HB carbon fixation pathway”

Reviewer's comment: p. 11: “Subsequently, the heterotrophic anaerobic archaea would acquire enzymes necessary for a 4HB or 3HP/4HB carbon fixation pathway and secure the ability to grow not only heterotrophically, but also chemolithoautotrophically, as seen in A3”: Fig. 4A3 shows succinyl-CoA carboxylation to 2-oxoglutarate. Is any archaeon with the 3HP/4HB or DC/4HB cycle known to use this reaction?

Fig. 4A2: what is “incomplete 4HB bypath”?

Reply: In the previous manuscript, DC/4HB pathway and 3HP/4HB pathway were not shown in Fig. 4A3. We added DC/4HB and 3HP/4HB pathway in Fig. 4A3. *Thermoproteus tenax* and *Acidianus hospitalis* possess the dicarboxylate/4-hydroxybutyrate (DC/4HB) pathway (DOI: 10.1371/journal.pone.0024222) and 3-hydroxypropionate/4-hydroxybutyrate (3HP/4HB) pathway (DOI: 10.1007/s00792-011-0379-y), respectively.

In Fig. 4A2, we revised “incomplete 4HB bypath” as “incomplete DC/4HB pathway”.

Reviewers' comments:

Reviewer #1 (Remarks to the Author):

[Editor's note: This reviewer provided no further comments to the authors.]

Reviewer #2 (Remarks to the Author):

The authors have improved their manuscript and addressed many of my previous comments. In particular, i now understand the nature of the results, as well as the fact that the model is implemented under a steady state assumption, with production of metabolites matched to their consumption, ultimately via biomass production. Assuming steady-state and solving for fluxes is generally similar to 'flux balance analysis' (FBA), a standard framework in systems biological research, and all seems reasonable and now more easily understandable.

However, there are a few conceptual points that I raised in my first review, which the authors have not really addressed, and therefore remain. I continue to struggle with some of the arguments that are developed in the paper, and the conclusions they lead to. Again, i think this work can make for a nice contribution by illuminating kinetic constraints on early metabolic evolution, but i think it would really help the reader if the authors could further clarify their logic on these points. All comments are offered in the spirit of trying to help the authors make their arguments as accessible as possible.

All the best,
Rogier Braakman

Comments:

1. Dynamics of coupled WL and rTCA cycle pathways in cells where acetate or methane outflux is included: in my first review I had asked whether including pathways for excreting acetyl-CoA (either as acetate or as methane) would resolve the kinetic conflict between the WL and rTCA cycle pathways, and allow them to co-occur in principle. Crucially, this is how the WL pathway works in the real world, where acetogens/methanogens direct flux of acetyl-CoA away from the TCA cycle as part of energy production. The authors did not really address this point. In their rebuttal letter, they responded:

"Reply: Thank you for the thoughtful comment. The kinetic network model basically includes outflux of ACOA as the biomass synthesis (Eq. B18 in the revised SI). On the other hand, we did not introduce an additional outflux of ACOA as an excretion pathway in the kinetic model simulations for methanogens (Figs. S5 b–e in the revised SI) and acetogens (Figs. S4 a, b in the revised SI). However, as shown in the SI figures 4 and 5, these organisms cannot have a complete rTCA cycle because of the lack of several enzymes for the rTCA cycle. This observation does not contradict with the kinetic hypothesis we proposed: a complete rTCA cycle is never observed in organisms because of its kinetic instability from the competition between the reductive and oxidative flux, as long as a carbon fixation pathway including the WL pathway yields ACOA influx into the rTCA cycle; unless the ACOA influx disappears, the gene for the enzyme working at the less active reaction on the rTCA cycle is lost or that of enzymes lacked on the less active part is not newly gained during evolution of LUCA."

I must admit this reasoning seems circular. Basically, the authors state that organisms with a WL pathway cannot have an rTCA pathway because they lack certain enzymes, which is a genomic observation, and therefore they did not include it in their model. Then, they argue that these organisms cannot have a complete rTCA cycle because of the kinetic conflict between the rTCA cycle and the WL pathway. But they have not shown this. For the genomic observations, it is not an established fact that the reason organism with a WL pathway lack rTCA enzymes is because of the kinetic conflict. That is your hypothesis, but there could be other reasons. For example, maybe organisms that have a WL pathway simply have no added benefit for also having the rTCA cycle.

If you want to conclusively argue that the kinetic conflict between the WL and rTCA cycle pathways is why they don't co-occur in the same organism, then I think you have to show this remains true in the presence of an outlet that can remove excess acetyl-CoA, whether as acetate or as methane. Indeed, the latter part of the quoted answer seems to indicate that the authors recognize that if flux acetyl-CoA is directed away from the TCA cycle the kinetic conflict might not occur...

Relatedly, in my first review I had also raised the point that experimentally observed acetyl-CoA concentrations appear to be about an order of magnitude lower than the results in this paper, and I asked how this affected the results/conclusions. Lower acetyl-CoA concentrations would appear to potentially lower (or perhaps even eliminate) the kinetic conflict between the WL and rTCA pathways.

The authors also did not really address this point, only pointing to results showing that when the influx of acetyl-CoA from the WL pathway into the rTCA cycle is higher, the kinetic conflict is greater. But this is exactly my point - if there was an outflux of acetyl-CoA to acetate or methane, there might not be a conflict between the WL and rTCA pathways.

2. Evolutionary consequences of the "kinetic hypothesis": i am still not quite understanding arguments on the evolutionary consequences of the kinetic conflict between the WL and rTCA cycle pathways. As mentioned in my previous review, the idea of evolution acting to eliminate or avoid this conflict implies there is another force acting to combine these two pathways. Organisms with one of these two pathways don't obviously need the second pathway if it doesn't provide a benefit, so I don't understand why the conflict would ever arise in the first place. In their rebuttal letter the authors replied as follows:

"Reply: Thank you for the thoughtful comments. We understood the possibility that there would be some other selective pressure acting in parallel that favors the combination of the rTCA cycle and WL pathway. At the beginning of the discussion, we revised the related sentence as follows. In the pioneering work by Braakman and Smith, a single connected redundant network consisting of a complete WL pathway and a full rTCA cycle was proposed as a more robust and plausible topology that (pre-) LUCA should possess on the basis of phylometabolic analysis. In contrast, the results obtained from our kinetic network model do not support their prediction on the coexistence of those pathway in one organism, unless other predominant selective pressure that favors the combination of those pathways exists in parallel."

I greatly appreciate the reference to our work, but this does not really address the point i raised, since it focuses on a different stage of evolution than I was referring to. I was referring to evolution

in the post-LUCA world, where cells and boundaries of individuality have become more firmly established. For example, in the introduction there is a sentence (bottom of page 2) "Consequently, the less active enzymes of the rTCA cycle due to this coexistence would be lost to evolution or enzymes lacked in the rTCA cycle would be difficult to be newly incorporated into the less active part of that." Similarly, the sections on deep-branching bacteria and archaea focus on the various ways in which the kinetic hypothesis is satisfied for different lineages. By definition, these writings examine evolution after the LUCA, since they focus on observations of diversity among extant lineages.

I understand your argument that the kinetic conflict (if it remains in the presence of a pathway for excreting acetate or methane) could prevent a lineage already possessing either the rTCA or the WL pathway from acquiring the other. However, why should such an acquisition ever happen in the first place? The framing of your argument implies that combining the rTCA and WL pathways would be beneficial, and that acquisition of the second pathway is only prevented by the kinetic conflict between the two pathways. But if there is no benefit to acquiring the second pathway, then there is no conflict to worry about.

I tried to think of examples of when having two pathways could be beneficial. One example I could think of was a scenario where organisms have two different CO₂-fixing pathways that they can switch between in two different environmental regimes, as has been proposed for the co-occurrence of the Calvin cycle and rTCA cycle in endosymbionts of tube worms at hydrothermal vents (Markert et al 2007 and subsequent papers). However, even in that scenario the two pathways would not be active at the same time, thus avoiding the kinetic conflict.

I would encourage the authors to further reflect on what might be the benefits for having and expressing both the WL and rTCA pathways in the extant biosphere, which they are currently implicitly assuming in much of their paper. If there is no such benefit then I don't clearly understand the consequences of the identified kinetic conflict between these pathways for post-LUCA evolution.

3. Questions of the LUCA and the distinction of pre-LUCA and post-LUCA evolution: there is still some ambiguity about the assumptions the authors make regarding pre-LUCA evolution and how it may have been different from post-LUCA evolution. I should emphasize this continues to be an area of lively debate, but it is at the very least plausible that in the pre-LUCA world, cells had less clearly defined boundaries of individuality due to less sophisticated/precise molecular machinery and more leaky membranes. Indeed, this must have been the case for at least parts of pre-LUCA evolution as it reaches back all the way to the prebiotic chemistry that preceded cells. Hence, I think it would still be good if the authors reflected a bit more on how their kinetic hypothesis would play out in the pre-LUCA world and how that affects their conclusions regarding the LUCA. For example, if membranes were inherently more leaky and metabolites continually lost from cells, perhaps this would lower the kinetic conflict between the WL and rTCA cycle pathways.

More specifically, in two different places in the paper (in the introduction and in the discussion) the authors include the following sentences, indicating them as part of the motivation for their work in this paper:

"In the pioneering work by Braakman and Smith 5, a single connected redundant network consisting of a complete WL pathway and a full rTCA cycle was proposed as a more robust and plausible

topology that (pre-) LUCA should possess on the basis of phylometabolic analysis. In contrast, the results obtained from our kinetic network model do not support their prediction on the coexistence of those pathway in one organism"

I again greatly appreciate the reference and discussion of our work, and indeed I'm honored that it provided part of the motivation of this work. However, I feel I must point out that in our paper we did not predict the occurrence of the WL and rTCA cycle pathways within a single organism. We concluded that a connected WL+rTCA pathway was the most likely CO₂-fixing pathway in the LUCA for reasons related to the self-organization of prebiotic chemical systems, but our work explicitly left unresolved whether this network was complete within single cells or only at the ecosystem level during the emergence of (proto)biological systems leading up to the LUCA.

Please refer to paragraph 4 in the introduction of our paper, which includes a sentence: "To define the constraint of autotrophy we will use metabolic-flux balance analysis, and because we do not use it to model cellular-level mechanisms of either regulation or heredity, it does not distinguish among strictly defined autotrophic species, populations of diverse cells (or pre-cells) tightly linked by transfer of genes and metabolites [11], [12], or syntrophic ecosystems."

Please also refer to the sub-section 'The ancestral carbon fixation phenotype' in the Results and Discussion section of our paper, which includes a sentence: "The less-clear separation of bacteria and archaea near the root in Fig. 5, the correspondence of these branches to the reticulated domain of gene phylogenies [12], [14], [19], the need and character of the inserted root node, and the flexible interpretation of our carbon-fixation phenotypes as species or consortia, leave open the possibility that the earliest branches were stages of chemical evolution that preceded modern life [3]–[6].".

Relatedly, in the rebuttal letter the authors write:

"Reply: We understood that it was important to distinguish between pre-LUCA and post-LUCA evolution in the discussions of early evolution of metabolism. Below the mention about the prediction by Braakman and Smith (at the beginning of the discussion), we added the following sentences.

The inconsistency between their and our predictions should be due to whether or not the kinetic factors are taken into consideration. On the other hand ..."

This interpretation again depends on the extent to which one assumes that pre-LUCA evolution concerns evolution of distinct cellular lineages. As mentioned above, it is entirely possible that pre-LUCA evolution was inherently more of an ecosystem-level process, with leaky consortia of diverse cells tightly intertwined at the level of metabolism. If anything, your results would potentially appear to suggest that a scenario along those lines is more likely.

As a consequence, I admit that I still don't see an incompatibility between your results and ours. Of course, it is entirely up to you how you want to discuss your results and the conclusions you want to draw from them, but at minimum I would kindly request our arguments are represented correctly.

Reviewer #3 (Remarks to the Author):

The revised manuscript is much more clear and understandable. Although I do not share all the thoughts of the authors, I think that the manuscript presents an interesting discussion and should be published.

One small point:

I am not sure whether the word "smaller" is correctly used in page 5: " 5×10^{-7} times smaller" and " 1×10^5 smaller", especially in the second case ("impaired reductive flux was $\sim 1 \times 10^5$ times smaller". For me, 10-5times smaller is 100000 times bigger. What is meant for the smaller impaired flux, I do not know. Please be more specific.

Reply to the comments by Reviewer #2

Dear Dr. Rogier Braakman,

We appreciate your thoughtful comments Reviewer's comments and our responses are listed below.

Reviewer's comment: 1. Dynamics of coupled WL and rTCA cycle pathways in cells where acetate or methane outflux is included: in my first review I had asked whether including pathways for excreting acetyl-CoA (either as acetate or as methane) would resolve the kinetic conflict between the WL and rTCA cycle pathways, and allow them to co-occur in principle. Crucially, this is how the WL pathway works in the real world, where acetogens/methanogens direct flux of acetyl-CoA away from the TCA cycle as part of energy production. The authors did not really address this point. In their rebuttal letter, they responded:

"Reply: Thank you for the thoughtful comment. The kinetic network model basically includes outflux of ACOA as the biomass synthesis (Eq. B18 in the revised SI). On the other hand, we did not introduce an additional outflux of ACOA as an excretion pathway in the kinetic model simulations for methanogens (Figs. S5 b–e in the revised SI) and acetogens (Figs. S4 a, b in the revised SI). However, as shown in the SI figures 4 and 5, these organisms cannot have a complete rTCA cycle because of the lack of several enzymes for the rTCA cycle. This observation does not contradict with the kinetic hypothesis we proposed: a complete rTCA cycle is never observed in organisms because of its kinetic instability from the competition between the reductive and oxidative flux, as long as a carbon fixation pathway including the WL pathway yields ACOA influx into the rTCA cycle; unless the ACOA influx disappears, the gene for the enzyme working at the less active reaction on the rTCA cycle is lost or that of enzymes lacked on the less active part is not newly gained during evolution of LUCA."

I must admit this reasoning seems circular. Basically, the authors state that organisms with a WL pathway cannot have an rTCA pathway because they lack certain enzymes, which is a genomic observation, and therefore they did not include it in their model. Then, they argue that these organisms cannot have a complete rTCA cycle because of the kinetic conflict between the rTCA cycle and the WL pathway. But they have not shown this. For the genomic observations, it is not an established fact that the reason organism with a WL pathway lack rTCA enzymes is because of the kinetic conflict. That is your hypothesis, but there could be other reasons. For example, maybe organisms that have a WL pathway simply have no added benefit for also having the rTCA cycle.

If you want to conclusively argue that the kinetic conflict between the WL and rTCA cycle pathways is why they don't co-occur in the same organism, then I think you have to show this remains true in the presence of an outlet that can remove excess acetyl-CoA, whether as acetate or as methane. Indeed, the latter part of the quoted answer seems to indicate that the authors recognize that if flux acetyl-CoA is directed away from the TCA cycle the kinetic conflict might not occur...

Relatedly, in my first review I had also raised the point that experimentally observed acetyl-CoA concentrations appear to be about an order of magnitude lower than the results in this paper, and I asked how this affected the results/conclusions. Lower acetyl-CoA concentrations would appear to potentially lower (or perhaps even eliminate) the kinetic conflict between the WL and rTCA pathways.

The authors also did not really address this point, only pointing to results showing that when the influx of acetyl-CoA from the WL pathway into the rTCA cycle is higher, the kinetic conflict is greater. But this is exactly my point - if there was an outflux of acetyl-CoA to acetate or methane, there might not be a conflict between the WL and rTCA pathways.

Reply: As you pointed out, the lower the ACOA influx is, the lower the kinetic conflict is. However, we conformed that an ACOA influx reduced the reductive flux from AKG to ACOA and OAA even if the ACOA influx was much smaller than the normal reductive flux from OAA to AKG, indicating that the conflict between rTCA cycle and ACOA influx. This result was added in the SI (see Fig. S4).

Reviewer's comment: 2. Evolutionary consequences of the "kinetic hypothesis": i am still not quite understanding arguments on the evolutionary consequences of the kinetic conflict between the WL and rTCA cycle pathways. As mentioned in my previous review, the idea of evolution acting to eliminate or avoid this conflict implies there is another force acting to combine these two pathways. Organisms with one of these two pathways don't obviously need the second pathway if it doesn't provide a benefit, so I don't understand why the conflict would ever arise in the first place. In their rebuttal letter the authors replied as follows:

"Reply: Thank you for the thoughtful comments. We understood the possibility that there would be some other selective pressure acting in parallel that favors the combination of the rTCA cycle and WL pathway. At the beginning of the discussion, we revised the related sentence as follows.

In the pioneering work by Braakman and Smith, a single connected redundant network consisting of a complete WL pathway and a full rTCA cycle was proposed as a more robust and plausible topology that (pre-) LUCA should possess on the basis of phylometabolic analysis. In contrast, the results obtained from our kinetic network model do not support their prediction on the coexistence of those

pathway in one organism, unless other predominant selective pressure that favors the combination of those pathways exists in parallel."

I greatly appreciate the reference to our work, but this does not really address the point I raised, since it focuses on a different stage of evolution than I was referring to. I was referring to evolution in the post-LUCA world, where cells and boundaries of individuality have become more firmly established. For example, in the introduction there is a sentence (bottom of page 2) "Consequently, the less active enzymes of the rTCA cycle due to this coexistence would be lost to evolution or enzymes lacked in the rTCA cycle would be difficult to be newly incorporated into the less active part of that." Similarly, the sections on deep-branching bacteria and archaea focus on the various ways in which the kinetic hypothesis is satisfied for different lineages. By definition, these writings examine evolution after the LUCA, since they focus on observations of diversity among extant lineages.

I understand your argument that the kinetic conflict (if it remains in the presence of a pathway for excreting acetate or methane) could prevent a lineage already possessing either the rTCA or the WL pathway from acquiring the other. However, why should such an acquisition ever happen in the first place? The framing of your argument implies that combining the rTCA and WL pathways would be beneficial, and that acquisition of the second pathway is only prevented by the kinetic conflict between the two pathways. But if there is no benefit to acquiring the second pathway, then there is no conflict to worry about.

I tried to think of examples of when having two pathways could be beneficial. One example I could think of was a scenario where organisms have two different CO₂-fixing pathways that they can switch between in two different environmental regimes, as has been proposed for the co-occurrence of the Calvin cycle and rTCA cycle in endosymbionts of tube worms at hydrothermal vents (Markert et al 2007 and subsequent papers). However, even in that scenario the two pathways would not be active at the same time, thus avoiding the kinetic conflict.

I would encourage the authors to further reflect on what might be the benefits for having _and expressing_ both the WL and rTCA pathways _in the extant biosphere_, which they are currently implicitly assuming in much of their paper. If there is no such benefit then I don't clearly understand the consequences of the identified kinetic conflict between these pathways for _post-LUCA evolution_.

Reply: Thank you for the thoughtful comments. As you pointed out, we did not argue benefits for having both the WL and rTCA pathways in one organism. On the other hand, our results provided the

non-trivial kinetic reason why organisms having both the WL and rTCA pathways did not exist in the present and hardly appeared during evolution unless other predominant selective pressure that favored their combination existed.

Reviewer's comment: 3. Questions of the LUCA and the distinction of pre-LUCA and post-LUCA evolution: there is still some ambiguity about the assumptions the authors make regarding pre-LUCA evolution and how it may have been different from post-LUCA evolution. I should emphasize this continues to be an area of lively debate, but it is at the very least plausible that in the pre-LUCA world, cells had less clearly defined boundaries of individuality due to less sophisticated/precise molecular machinery and more leaky membranes. Indeed, this must have been the case for at least parts of pre-LUCA evolution as it reaches back all the way to the prebiotic chemistry that preceded cells. Hence, I think it would still be good if the authors reflected a bit more on how their kinetic hypothesis would play out in the pre-LUCA world and how that affects their conclusions regarding the LUCA. For example, if membranes were inherently more leaky and metabolites continually lost from cells, perhaps this would lower the kinetic conflict between the WL and rTCA cycle pathways.

More specifically, in two different places in the paper (in the introduction and in the discussion) the authors include the following sentences, indicating them as part of the motivation for their work in this paper:

"In the pioneering work by Braakman and Smith 5, a single connected redundant network consisting of a complete WL pathway and a full rTCA cycle was proposed as a more robust and plausible topology that (pre-) LUCA should possess on the basis of phylometabolic analysis. In contrast, the results obtained from our kinetic network model do not support their prediction on the coexistence of those pathway in one organism"

I again greatly appreciate the reference and discussion of our work, and indeed I'm honored that it provided part of the motivation of this work. However, I feel I must point out that in our paper we did not predict the occurrence of the WL and rTCA cycle pathways within a single organism. We concluded that a connected WL+rTCA pathway was the most likely CO₂-fixing pathway in the LUCA for reasons related to the self-organization of prebiotic chemical systems, but our work explicitly left unresolved whether this network was complete within single cells or only at the ecosystem level during the emergence of (proto)biological systems leading up to the LUCA.

Please refer to paragraph 4 in the introduction of our paper, which includes a sentence: "To define the

constraint of autotrophy we will use metabolic-flux balance analysis, and because we do not use it to model cellular-level mechanisms of either regulation or heredity, it does not distinguish among strictly defined autotrophic species, populations of diverse cells (or pre-cells) tightly linked by transfer of genes and metabolites [11], [12], or syntrophic ecosystems."

Please also refer to the sub-section 'The ancestral carbon fixation phenotype' in the Results and Discussion section of our paper, which includes a sentence: "The less-clear separation of bacteria and archaea near the root in Fig. 5, the correspondence of these branches to the reticulated domain of gene phylogenies [12], [14], [19], the need and character of the inserted root node, and the flexible interpretation of our carbon-fixation phenotypes as species or consortia, leave open the possibility that the earliest branches were stages of chemical evolution that preceded modern life [3]–[6]."

Relatedly, in the rebuttal letter the authors write:

"Reply: We understood that it was important to distinguish between pre-LUCA and post-LUCA evolution in the discussions of early evolution of metabolism. Below the mention about the prediction by Braakman and Smith (at the beginning of the discussion), we added the following sentences. The inconsistency between their and our predictions should be due to whether or not the kinetic factors are taken into consideration. On the other hand ..."

This interpretation again depends on the extent to which one assumes that pre-LUCA evolution concerns evolution of distinct cellular lineages. As mentioned above, it is entirely possible that pre-LUCA evolution was inherently more of an ecosystem-level process, with leaky consortia of diverse cells tightly intertwined at the level of metabolism. If anything, your results would potentially appear to suggest that a scenario along those lines is more likely.

As a consequence, I admit that I still don't see an incompatibility between your results and ours. Of course, it is entirely up to you how you want to discuss your results and the conclusions you want to draw from them, but at minimum I would kindly request our arguments are represented correctly.

Reply: Thank you again for explaining the point of your work. In contrast to your work including (proto)biological systems leading up to the LUCA, carbon metabolism pathways on the protobiological systems are out of scope in our present work because the concentrations of chemical species (listed in Table A1), ionic strength, and pH are unknown. Thus, we did not expand the argument with the kinetic network model to prebiotic chemical systems. On the other hand, we would like to

refer your arguments correctly as much as possible, we revised the explanation on your arguments at the beginning of the discussion (and also in the introduction).

In the pioneering work by Braakman and Smith, a single connected redundant network consisting of a complete WL pathway and a full rTCA cycle was proposed as a more robust and plausible topology that LUCA or (proto) biological systems leading up to the LUCA possibly possess on the basis of phylometabolic analysis.

The above-mentioned sentence is followed by our statement shown below, which does not refer to the protobiological systems.

In contrast, the results obtained from our kinetic network model do not support the coexistence of those pathway in one organism, unless a more dominant and selective pressure that favors the combination of those pathways exists in parallel.

Reply to the comments by Reviewer #3

We appreciate your careful reading of the manuscript. Reviewer's comments and our responses are listed below.

Reviewer's comment: I am not sure whether the word "smaller" is correctly used in page 5: " 5×10^{-7} times smaller" and " 1×10^5 smaller", especially in the second case ("impaired reductive flux was $\sim 1 \times 10^5$ times smaller". For me, 10-5times smaller is 100000 times bigger. What is meant for the smaller impaired flux, I do not know. Please be more specific.

Reply: Thank you for the crucial suggestions. We revised the unsuitable expression " 5×10^{-7} times smaller" as " 5×10^7 times smaller".

REVIEWERS' COMMENTS:

Reviewer #2 (Remarks to the Author):

As said before, this paper makes an interesting contribution on the topic of kinetic constraints on early metabolic evolution. It is appropriate to proceed with publication. The new SI figure that systematically explores the kinetic conflict between the reductive acetyl-CoA pathway and rTCA cycle as a function of acetyl-CoA influx is a nice addition. I still don't quite understand some of the conclusions or agree with all the logic, but that is ok. Comments in previous reviews were in any case not intended as some kind of requirement for publication, they were intended to offer the author an opportunity to clarify on some points that could lead to confusion. But it is of course up to the authors how they want to present their work.

I do have two minor comments remaining, which are similar to my previous review:

In the introduction (last paragraph before results), in reference to our previously published work (Braakman & Smith, 2012) it still says that "our kinetic results do not support THEIR PREDICTED coexistence of these two pathways IN ONE ORGANISM". As mentioned, that is not what we predicted in our paper. If you just removed 'THEIR PREDICTED' i think it would be fine.

Similarly, in the discussion (first paragraph) it says "In contrast, the results obtained from our kinetic network model do not support the coexistence of those pathway IN ONE ORGANISM, unless a more dominant and selective pressure that favors the combination of those pathways exists in parallel. THE INCONSISTENCY BETWEEN THEIR PROPOSAL AND OURS should be due to whether the kinetic factors are taken into consideration". As mentioned, we did not propose or predict that these pathways should co-occur in one organism. Hence, another resolution that could be mentioned would be whether one assumes that the LUCA consisted of a single organism or a diverse consortia of metabolically coupled cells. As mentioned, we explicitly left the latter open as a possibility.

All the best,
Rogier Braakman

Reply to the comments by Reviewer #2

Dear Dr. Rogier Braakman,

We appreciate your careful checking and helpful suggestions. Our responses are listed below.

Reviewer's comment: In the introduction (last paragraph before results), in reference to our previously published work (Braakman & Smith, 2012) it still says that "our kinetic results do not support THEIR PREDICTED coexistence of these two pathways IN ONE ORGANISM". As mentioned, that is not what we predicted in our paper. If you just removed 'THEIR PREDICTED' i think it would be fine.

Reply: According to your suggestion, we revised "their predicted coexistence" as "the coexistence".

Reviewer's comment: Similarly, in the discussion (first paragraph) it says "In contrast, the results obtained from our kinetic network model do not support the coexistence of those pathway IN ONE ORGANISM, unless a more dominant and selective pressure that favors the combination of those pathways exists in parallel. THE INCONSISTENCY BETWEEN THEIR PROPOSAL AND OURS should be due to whether the kinetic factors are taken into consideration". As mentioned, we did not propose or predict that these pathways should co-occur in one organism. Hence, another resolution that could be mentioned would be whether one assumes that the LUCA consisted of a single organism or a diverse consortia of metabolically coupled cells. As mentioned, we explicitly left the latter open as a possibility.

Reply: According to your suggestions, we revised as follows:

The key difference between their work and ours is whether the kinetic factors are taken into consideration.

Due to this, we removed the following sentence.

On the other hand, their arguments on post-LUCA evolution does not seem to be inconsistent with our kinetic hypothesis.